# Fast Tensor Completion via Approximate Richardson Iteration

**Mehrdad Ghadiri** [1]  **Matthew Fahrbach** [2]  **Yunbum Kook** [3]  **Ali Jadbabaie** [1]

## Abstract

We study tensor completion (TC) through the lens of low-rank tensor decomposition (TD). Many TD algorithms use fast alternating minimization methods to solve *highly structured* linear regression problems at each step (e.g., for CP, Tucker, and tensor-train decompositions). However, such algebraic structure is often lost in TC regression problems, making direct extensions unclear. This work proposes a novel *lifting* method for approximately solving TC regression problems using structured TD regression algorithms as blackbox subroutines, enabling sublinear-time methods. We analyze the convergence rate of our approximate Richardson iteration-based algorithm, and our empirical study shows that it can be 100x faster than direct methods for CP completion on real-world tensors.

## 1. Introduction

*Tensor completion* (TC) is the higher-order generalization of matrix completion. It has many applications across data mining, machine learning, signal processing, and statistics (see Song et al. (2019) for a detailed survey of applications). In the TC problem, we are given a partially observed tensor as input (i.e., only a subset of its entries is known), and the goal is to impute the missing values. Under low-rank and other statistical assumptions, we can recover the missing values by minimizing a loss function over only the observed entries.

Consider a tensor $\mathcal{X} \in \mathbb{R}^{I_1 \times \cdots \times I_N}$ with a subset of observed indices $\Omega \subseteq [I_1] \times \cdots \times [I_N]$. The general TC problem is

$$\min_{\boldsymbol{\theta}} \mathcal{L}_{\Omega}(\widehat{\mathcal{X}}(\boldsymbol{\theta}), \mathcal{X}) + \mathcal{R}(\boldsymbol{\theta}), \quad (1)$$

where $\boldsymbol{\theta}$ are the learnable parameters, $\widehat{\mathcal{X}}(\boldsymbol{\theta}) \in \mathbb{R}^{I_1 \times \cdots \times I_N}$ is the reconstructed tensor, $\mathcal{L}_{\Omega}$ is the loss function defining the error between $\widehat{\mathcal{X}}(\boldsymbol{\theta})$ and $\mathcal{X}$ on entries in $\Omega$, and $\mathcal{R}(\boldsymbol{\theta})$ is the

regularization term. For brevity, we write $\widehat{\mathcal{X}}$ without explicit dependence on $\boldsymbol{\theta}$. Rank constraints can be incorporated into (1) by including appropriate penalty terms in $\mathcal{R}(\boldsymbol{\theta})$, e.g., by using $\ell_1$ or $\ell_2$ regularization terms to reduce the effective dimension (Fahrbach et al., 2022).

This work focuses on the sum of squared errors

$$\mathcal{L}_{\Omega}(\widehat{\mathcal{X}}, \mathcal{X}) = \sum_{(i_1, \ldots, i_N) \in \Omega} (\widehat{x}_{i_1 \ldots i_N} - x_{i_1 \ldots i_N})^2. \quad (2)$$

To introduce our approach, we use a running example where $\widehat{\mathcal{X}} = \mathcal{G} \times_1 \mathbf{A}^{(1)} \times_2 \mathbf{A}^{(2)} \cdots \times_N \mathbf{A}^{(N)}$ is a *low-rank Tucker decomposition* with core tensor $\mathcal{G} \in \mathbb{R}^{R_1 \times \cdots \times R_N}$ and factor matrices $\mathbf{A}^{(n)} \in \mathbb{R}^{I_n \times R_n}$. We do not focus on regularization in this paper, but our approach extends to regularized problems, provided there exists an algorithm that accelerates the corresponding structured and regularized objectives. For example, Fahrbach et al. (2022) present subquadratic-time algorithms for Kronecker ridge regression (i.e., $\ell_2$ regularization).

In general, TC is a nonconvex and NP-hard problem (Hillar & Lim, 2013). The special case where $\Omega = [I_1] \times \cdots \times [I_N]$ is the widely studied *tensor decomposition* (TD) problem, for which *alternating least squares* (ALS) methods are often used to compute $\mathcal{G}, \mathbf{A}^{(1)}, \ldots, \mathbf{A}^{(N)}$.

Each ALS step fixes all but one of $\mathcal{G}, \mathbf{A}^{(1)}, \ldots, \mathbf{A}^{(N)}$, and optimizes the unfixed component. In the case of TD, each ALS step is a *highly structured* linear regression problem. For example, in the core tensor update for $\mathcal{G}$, ALS solves

$$\min_{\mathcal{G}' \in \mathbb{R}^{R_1 \times \cdots \times R_N}} \left\| \left( \bigotimes_{n=1}^{N} \mathbf{A}^{(n)} \right) \mathrm{vec}(\mathcal{G}') - \mathrm{vec}(\mathcal{X}) \right\|_2, \quad (3)$$

where $\otimes$ is the *Kronecker product* and $\mathrm{vec}(\mathcal{G}') \in \mathbb{R}^R$ is the flattened version of tensor $\mathcal{G}'$, with $R := \prod_{n=1}^{N} R_n$.

Diao et al. (2019) and Fahrbach et al. (2022) recently exploited the Kronecker product structure in (3) to give algorithms with running times that are sublinear in the size of the full tensor $I := \prod_{n=1}^{N} I_n$. These approaches use *leverage score sampling* to approximately solve (3).

In TC, however, only a subset of observations appear in the loss function. Letting $\mathbf{A} = \bigotimes_{n=1}^{N} \mathbf{A}^{(n)}$ and $\mathbf{b} = \mathrm{vec}(\mathcal{X})$,

*Equal contribution [1]MIT [2]Google Research [3]Georgia Tech. Correspondence to: Mehrdad Ghadiri <mehrdadg@mit.edu>.

*Proceedings of the 42nd International Conference on Machine Learning*, Vancouver, Canada. PMLR 267, 2025. Copyright 2025 by the author(s).

the core tensor update becomes

$$\text{vec}(\mathcal{G}) \leftarrow \underset{\mathbf{x} \in \mathbb{R}^R}{\arg\min} \|\mathbf{A}_\Omega \mathbf{x} - \mathbf{b}_\Omega\|_2, \qquad (4)$$

where $\mathbf{A}_\Omega$ is a submatrix of $\mathbf{A}$ whose rows correspond to the indices in $\Omega$. Observe that the Kronecker product structure in (3) *no longer exists* for $\mathbf{A}_\Omega$ in (4), hence fast TD methods do not immediately extend to $\Omega$-masked TC versions.

A natural idea to overcome the lack of structure in the $\Omega$-masked updates is to lift (4) to a higher-dimensional problem by introducing variables $\mathbf{b}_{\overline{\Omega}}$, where $\overline{\Omega}$ is the complement of $\Omega$, i.e., letting the unobserved entries in $\mathcal{X}$ be *free variables*. This is a convex problem with much of the same structure as the design matrix in the full TD problem. Further, it gives the same solution as the TC update in (4):

$$(\mathbf{x}^*, \mathbf{b}_{\overline{\Omega}}^*) = \underset{\mathbf{x}, \mathbf{b}_{\overline{\Omega}}}{\arg\min} \|\mathbf{A}\mathbf{x} - \mathbf{b}\|_2. \qquad (5)$$

To our knowledge, such ideas date back to Healy & Westmacott (1956) in the experimental design and causal inference literature. However, it is not clear a priori that lifting is helpful for *computational efficiency*—it restores the structure of the design matrix, but the new problem (5) is larger and higher dimensional.

This work proposes solving (5) with a two-step procedure called *mini-ALS*. Given a vector $\mathbf{x}^{(k)}$ corresponding to an iterate of a block of variables in the low-rank decomposition, it repeats the following:

1. Set $\mathbf{b}_{\overline{\Omega}}^{(k)} \leftarrow \mathbf{A}_{\overline{\Omega}} \mathbf{x}^{(k)}$ and $\mathbf{b}_\Omega^{(k)} \leftarrow \mathbf{b}_\Omega$     // lift

2. Set $\mathbf{x}^{(k+1)} \leftarrow \arg\min_{\mathbf{x}} \|\mathbf{A}\mathbf{x} - \mathbf{b}^{(k)}\|_2$     // solve

Mini-ALS iterations can still be expensive since $\mathbf{A} \in \mathbb{R}^{I \times R}$ is a tall-and-skinny matrix (i.e., $I \gg R$). However, step two is a structured ALS for TD, so we propose solving it *approximately* with row sampling, allowing us to tap into a rich line of work on leverage score sampling for CP decomposition (Cheng et al., 2016; Larsen & Kolda, 2022; Bharadwaj et al., 2023), Tucker decomposition (Diao et al., 2019; Fahrbach et al., 2022), and tensor-train decomposition (Bharadwaj et al., 2024).

Returning to the running Tucker completion example, updating the core tensor in step two of mini-ALS via (3) only requires sampling $\tilde{O}(R)$ rows of $\mathbf{A}$, which is a substantially *smaller* problem. Further, we can compute $\mathbf{b}^{(k)}$ *lazily*, i.e., only the entries corresponding to sampled rows. We call our lifted iterative method *approximate-mini-ALS*.

## 1.1. Our Contributions

We summarize our main contributions as follows:

- In Section 3, we propose using the mini-ALS algorithm for each step of ALS for TC. We show that it simulates the *preconditioned Richardson iteration* (Lemma 3.5). Our main theoretical contribution is proving that the second step of a mini-ALS iteration can be performed *approximately*. We quantify how small the approximation error must be for approximate-mini-ALS to converge at the same rate as the Richardson iteration (Theorem 3.7). This lets us to extend a recent line of work on leverage score sampling-based TD ALS algorithms to the TC setting.

- In Section 4, we use state-of-the-art TD ALS algorithms for CP, Tucker, and tensor-train decompositions as *blackbox subroutines* in Algorithm 1 to obtain novel sampling-based TC algorithms. Hence, our lifting approach for TC also benefits from future TD algorithmic improvements.

- In Section 5, we show that leverage score sampling is an effective method for solving large structured regression problems via the *coupled matrix problem*. Then we compare the empirical performance of our lifted algorithm to direct methods for *low-rank CP completion* on synthetic and real-world tensors. We observe that mini-ALS can be orders of magnitude faster than direct ALS methods, while achieving comparable reconstruction errors. Finally, we propose an *accelerated* version with adaptive step sizes that extrapolates the trajectory of the iterates $\mathbf{x}^{(k)}$ and converges in fewer iterations.

## 1.2. Related Work

**CP completion.** Tomasi & Bro (2005) proposed an ALS algorithm for CP completion that repeats the following two-step process: (1) fill in the missing values using the current CP decomposition, and (2) update one factor matrix. Their algorithm is equivalent to running *one iteration* of mini-ALS in each step of ALS. As Tomasi & Bro (2005) discuss, this can lead to slower convergence and an increased likelihood of converging to suboptimal local minima because of errors introduced by the imputed missing values.

In contrast, approximate-mini-ALS runs *until convergence* in each step of ALS. By doing this, we establish a connection to the Richardson iteration and build on its convergence guarantees. Further, Tomasi & Bro (2005) explicitly fill in *all missing values* using the current decomposition, whereas we only impute missing values *required by row sampling*, allowing us to achieve sublinear-time updates in the size of the tensor. In general, iteratively fitting to imputed missing values falls under the umbrella of *expectation-maximization* (EM) algorithms (Little & Rubin, 2019, Chapter 8).

**Statistical assumptions.** Similar to minimizing the nuclear norm for matrix completion (Fazel, 2002; Candes &

Recht, 2012), a line of research in noisy TC proposes minimizing a convex relaxation of rank and identifies statistical assumptions under which the problem is recoverable (Barak & Moitra, 2016). Two standard assumptions are *incoherence* and the *missing-at-random* assumption. In Section 3, we discuss how these assumptions provide a bound on the number of steps required for mini-ALS. Alternating minimization approaches have also been applied in this line of noisy TC research (Jain & Oh, 2014; Liu & Moitra, 2020). It is also consistently observed that methods based on TD and tensor unfoldings are more practical and computationally efficient (Acar et al., 2011; Montanari & Sun, 2018; Filipović & Jukić, 2015; Shah & Yu, 2019; 2023).

## 2. Preliminaries

**Notation.** The *order* of a tensor is the number of its dimensions $N$. We denote scalars by normal lowercase letters $x \in \mathbb{R}$, vectors by boldface lowercase letters $\mathbf{x} \in \mathbb{R}^n$, matrices by boldface uppercase letters $\mathbf{X} \in \mathbb{R}^{m \times n}$, and higher-order tensors by boldface script letters $\mathcal{X} \in \mathbb{R}^{I_1 \times \cdots \times I_N}$. We use normal uppercase letters for the size of an index set, e.g., $[N] = \{1, 2, \ldots, N\}$. We define $I_{\neq k} := I/I_k$ for $k \in [N]$ and similarly $R_{\neq k := R/R_k}$. We denote the $i$-th entry of $\mathbf{x}$ by $x_i$, the $(i, j)$-th entry of $\mathbf{X}$ by $x_{ij}$, and the $(i, j, k)$-th entry of a third-order tensor $\mathcal{X}$ by $x_{ijk}$.

**Linear algebra.** A symmetric matrix $\mathbf{A} \in \mathbb{R}^{n \times n}$ is *positive semi-definite* (PSD) if $\mathbf{v}^\top \mathbf{A} \mathbf{v} \geq 0$ for any $\mathbf{v} \in \mathbb{R}^n$. For two symmetric matrices $\mathbf{A}$, $\mathbf{B}$, we use $\mathbf{A} \preccurlyeq \mathbf{B}$ to indicate that $\mathbf{B} - \mathbf{A}$ is PSD. For a PSD matrix $\mathbf{M} \in \mathbb{R}^{n \times n}$ and vector $\mathbf{v} \in \mathbb{R}^n$, we define $\|\mathbf{v}\|_{\mathbf{M}} := (\mathbf{v}^\top \mathbf{M} \mathbf{v})^{1/2}$. For $\mathbf{A} \in \mathbb{R}^{m \times n}$, $\mathbf{b} \in \mathbb{R}^m$, and $\Omega \subseteq [m]$, we use $\mathbf{A}_\Omega \in \mathbb{R}^{|\Omega| \times n}$ and $\mathbf{b}_\Omega \in \mathbb{R}^{|\Omega|}$ to denote the submatrix and subvector with rows indexed by $\Omega$. We let $\otimes$ denote the Kronecker product and $\odot$ denote the Khatri–Rao product.

**Tensor algebra.** The fibers of a tensor are the vectors obtained by fixing all but one index, e.g., if $\mathcal{X} \in \mathbb{R}^{I \times J \times K}$, the column, row and tube fibers are $\mathbf{x}_{:jk}$, $\mathbf{x}_{i:k}$, and $\mathbf{x}_{ij:}$, respectively. The *mode-$n$ unfolding* of a tensor $\mathcal{X} \in \mathbb{R}^{I_1 \times \cdots \times I_N}$ is the matrix $\mathbf{X}_{(n)} \in \mathbb{R}^{I_n \times (I_1 \cdots I_{n-1} I_{n+1} \cdots I_N)}$ that arranges the mode-$n$ fibers of $\mathcal{X}$ as rows of $\mathbf{X}_{(n)}$ sorted lexicographically by index. The *vectorization* $\mathrm{vec}(\mathcal{X}) \in \mathbb{R}^{I_1 \cdots I_N}$ of $\mathcal{X}$ stacks the elements of $\mathcal{X}$ lexicographically by index.

For $n \in [N]$, we denote the *mode-$n$ product* of a tensor $\mathcal{X} \in \mathbb{R}^{I_1 \times \cdots \times I_N}$ and matrix $\mathbf{A} \in \mathbb{R}^{J \times I_n}$ by $\mathcal{Y} = \mathcal{X} \times_n \mathbf{A}$, where $\mathcal{Y} \in \mathbb{R}^{I_1 \times \cdots \times I_{n-1} \times J \times I_{n+1} \times \cdots \times I_N}$. This operation multiplies each mode-$n$ fiber of $\mathcal{X}$ by $\mathbf{A}$, and is expressed elementwise as

$$(\mathcal{X} \times_n \mathbf{A})_{i_1 \ldots i_{n-1} j i_{n+1} \ldots i_N} = \sum_{i_n=1}^{I_n} x_{i_1 i_2 \ldots i_N} a_{j i_n}.$$

The inner product of two tensors $\mathcal{X}, \mathcal{Y} \in \mathbb{R}^{I_1 \times \cdots \times I_N}$ is

$$\langle \mathcal{X}, \mathcal{Y} \rangle = \sum_{i_1=1}^{I_1} \sum_{i_2=1}^{I_2} \cdots \sum_{i_N=1}^{I_N} x_{i_1 i_2 \ldots i_N} y_{i_1 i_2 \ldots i_N}.$$

The Frobenius norm of a tensor $\mathcal{X}$ is $\|\mathcal{X}\|_{\mathrm{F}} = \sqrt{\langle \mathcal{X}, \mathcal{X} \rangle}$.

### 2.1. Tensor Decompositions

The tensor decompositions of $\mathcal{X} \in \mathbb{R}^{I_1 \times \cdots \times I_N}$ below can be seen as higher-order analogs of low-rank matrix factorization. We direct the reader to Kolda & Bader (2009) for a comprehensive survey on this topic.

**CP decomposition.** A rank-$R$ *CP decomposition* represents $\mathcal{X}$ with $\boldsymbol{\lambda} \in \mathbb{R}^R_{\geq 0}$ and $N$ factors $\mathbf{A}^{(n)} \in \mathbb{R}^{I_n \times R}$, for $n \in [N]$, where each column of $\mathbf{A}^{(n)}$ has unit norm. The reconstructed tensor $\widehat{\mathcal{X}}$ is defined elementwise as:

$$\widehat{x}_{i_1 \ldots i_N} = \sum_{r=1}^R \lambda_r \, a_{i_1 r}^{(1)} \cdots a_{i_N r}^{(N)}.$$

**Tucker decomposition.** A rank-$\mathbf{r}$ *Tucker decomposition* represents $\mathcal{X}$ with a *core tensor* $\mathcal{G} \in \mathbb{R}^{R_1 \times \cdots \times R_N}$ and $N$ *factor matrices* $\mathbf{A}^{(n)} \in \mathbb{R}^{I_n \times R_n}$, for $n \in [N]$, where $\mathbf{r} = (R_1, \ldots, R_N)$ is the multilinear rank (Ghadiri et al., 2023a). The reconstructed tensor $\widehat{\mathcal{X}} = \mathcal{G} \times_1 \mathbf{A}^{(1)} \times_2 \cdots \times_N \mathbf{A}^{(N)}$ is defined elementwise as:

$$\widehat{x}_{i_1 \ldots i_N} = \sum_{r_1=1}^{R_1} \cdots \sum_{r_N=1}^{R_N} g_{r_1 \ldots r_N} a_{i_1 r_1}^{(1)} \cdots a_{i_N r_N}^{(N)}.$$

**TT decomposition.** A rank-$\mathbf{r}$ *tensor train (TT) decomposition* (Oseledets, 2011) represents $\mathcal{X}$ with $N$ third-order *TT-cores* $\mathcal{A}^{(n)} \in \mathbb{R}^{R_{n-1} \times I_n \times R_n}$, for $n \in [N]$, using the convention $R_0 = R_N = 1$. The reconstructed tensor $\widehat{\mathcal{X}}$ is defined elementwise as:

$$\widehat{x}_{i_1 \ldots i_N} = \underbrace{\mathbf{A}_{:i_1:}^{(1)}}_{1 \times R_1} \underbrace{\mathbf{A}_{:i_2:}^{(2)}}_{R_1 \times R_2} \cdots \underbrace{\mathbf{A}_{:i_{N-1}:}^{(N-1)}}_{R_{N-2} \times R_{N-1}} \underbrace{\mathbf{A}_{:i_N:}^{(N)}}_{R_{N-1} \times 1}.$$

*Remark* 2.1. All three of these TDs are instances of the more general *tensor network* framework (see Appendix A.4).

### 2.2. ALS Formulations

*Alternating least squares* (ALS) methods are the gold standard for low-rank tensor decomposition, e.g., they are the first techniques mentioned in the MATLAB Tensor Toolbox (Bader & Kolda, 2023). ALS cyclically minimizes the original least-squares problem (1) with respect to one block of factor variables while keeping all others fixed. Repeating

this process converges to a nontrivial local optimum and reduces the original nonconvex problem to a series of (convex) linear regression problems in each step (see Appendix A.1).

Updating a block of variables in an ALS step of a TD problem is often a *highly structured* regression problem that can be solved very fast with problem-specific algorithms. We describe the induced structure of each ALS update for the tensor decomposition types above.

**CP factor matrix update ≡ Khatri–Rao regression.** In each ALS step for CP decomposition, all factor matrices are fixed except for one, say $\mathbf{A}^{(n)}$. ALS solves the following linear least-squares problem:

$$\mathbf{A}^{(n)} \leftarrow \underset{\mathbf{A} \in \mathbb{R}^{I_n \times R}}{\arg\min} \left\| \left( \bigodot_{i=1, i \neq n}^{N} \mathbf{A}^{(i)} \right) \mathbf{A}^\top - \mathbf{X}_{(n)}^\top \right\|_{\mathrm{F}}. \quad (6)$$

Then, we set $\lambda_r = \|\mathbf{a}_{:r}^{(n)}\|_2$ for each $r \in [R]$ and normalize the columns of $\mathbf{A}^{(n)}$. Each row of $\mathbf{A}^{(n)}$ can be optimized independently, so (6) solves $I_n$ linear regression problems where the design matrix is a *Khatri–Rao product*.

**Tucker core update ≡ Kronecker regression.** The core-tensor ALS update solves the following: for $\mathbf{A}^{(n)}$ fixed,

$$\mathcal{G} \leftarrow \underset{\mathcal{G}' \in \mathbb{R}^{R_1 \times \cdots \times R_N}}{\arg\min} \left\| \left( \bigotimes_{n=1}^{N} \mathbf{A}^{(n)} \right) \mathrm{vec}(\mathcal{G}') - \mathrm{vec}(\mathcal{X}) \right\|_2,$$

where the design matrix is a *Kronecker product* of the factors.

**Tucker factor update ≡ Kronecker-matrix regression.** When ALS updates $\mathbf{A}^{(n)}$ with all the other factor matrices and core tensor fixed, it solves:

$$\mathbf{A}^{(n)} \leftarrow \underset{\mathbf{A} \in \mathbb{R}^{I_n \times R_n}}{\arg\min} \left\| \left( \bigotimes_{i=1, i \neq n}^{N} \mathbf{A}^{(i)} \right) \mathbf{G}_{(n)}^\top \mathbf{A}^\top - \mathbf{X}_{(n)}^\top \right\|_{\mathrm{F}},$$

where $\mathbf{G}_{(n)}, \mathbf{X}_{(n)}$ are the mode-$n$ unfoldings of $\mathcal{G}$ and $\mathcal{X}$. This is equivalent to solving $I_n$ independent linear regression problems, where the design matrix is the product of a Kronecker product and another matrix. It can be viewed as solving $I_n$ instances of structured but *constrained linear regression* (Fahrbach et al., 2022).

**TT-core update ≡ Kronecker regression.** Given a TT decomposition $\{\mathcal{A}^{(n)}\}_{n=1}^{N}$ and $n \in [N]$, the *left chain* $\mathbf{A}_{<n} \in \mathbb{R}^{(I_1 \cdots I_{n-1}) \times R_{n-1}}$ and *right chain* $\mathbf{A}_{>n} \in \mathbb{R}^{R_n \times (I_{n+1} \cdots I_N)}$ are matrices that depend on the cores $\mathcal{A}^{(n')}$ for $n' < n$ and $n' > n$, respectively (see Appendix A.3 for details). When ALS updates $\mathcal{A}^{(n)}$ with all other TT-cores fixed, it solves

$$\mathcal{A}^{(n)} \leftarrow \underset{\mathcal{B} \in \mathbb{R}^{R_{n-1} \times I_n \times R_n}}{\arg\min} \left\| (\mathbf{A}_{<n} \otimes \mathbf{A}_{>n}^\top) \mathbf{B}_{(2)}^\top - \mathbf{X}_{(n)}^\top \right\|_{\mathrm{F}},$$

which is equivalent to solving $I_n$ Kronecker regression problems in $\mathbb{R}^{I_{\neq n}}$.

## 3. Approximate Richardson Iteration

We now present our main techniques for reducing tensor completion to tensor decomposition. When using ALS to solve a TC problem, we must efficiently solve least-squares problems

$$\min_{\mathbf{x}} \|\mathbf{Px} - \mathbf{q}\|_2.$$

The rows of the design matrix $\mathbf{P}$ correspond to the *subset of observations* in the TC problem, which means $\mathbf{P}$ does not necessarily have the structure of the design matrix in the full TD problem.

A direct approach is to compute the closed-form solution $\mathbf{x}^* = (\mathbf{P}^\top \mathbf{P})^{-1} \mathbf{P}^\top \mathbf{q}$, but computing $(\mathbf{P}^\top \mathbf{P})^{-1}$ is often impractical. Two techniques are commonly used to overcome this: (1) iterative methods and (2) row sampling. Iterative methods repeat the same relatively cheap per-step computation *many times* to approximate the original expensive computation. Row sampling methods (e.g., leverage score sampling) randomly pick rows of $\mathbf{P}$ and solve a least-squares problem on the sampled rows to obtain an approximate solution to the original problem with high probability. Directly computing leverage scores for a general $\mathbf{P}$, however, is *also prohibitively expensive* since it requires computing the same matrix $(\mathbf{P}^\top \mathbf{P})^{-1}$ (see Appendix A.2 for details).

We show that our *approximate-mini-ALS* method is a principled approach for tensor completion. In Section 3.1, we prove that lifting *restores the structure* of the full TD ALS update step, enabling fast least-squares methods for a larger but equivalent problem. In Section 3.2, we show that iteratively solving the lifted problem (i.e., mini-ALS) is connected to an iterative method called the *Richardson iteration* (Richardson, 1911), which we can also view as a matrix-splitting method. In other words, mini-ALS and the Richardson iteration with a certain preconditioner give the same sequence of iterates $\{\mathbf{x}^{(k)}\}_{k \geq 0}$. Lastly in Section 3.3, we prove novel convergence guarantees for *approximately* solving the lifted problem (i.e., for approximate-mini-ALS). This allows us to directly use fast leverage-score sampling algorithms for CP decomposition (Cheng et al., 2016; Larsen & Kolda, 2022; Bharadwaj et al., 2023), Tucker decomposition (Diao et al., 2019; Fahrbach et al., 2022), and TT decomposition (Bharadwaj et al., 2024) as blackbox subroutines. All missing proofs are deferred to Appendix B.

### 3.1. Lifting to a Structured Problem

Consider the linear regression problem with $\mathbf{P} \in \mathbb{R}^{|\Omega| \times R}$ and $\mathbf{q} \in \mathbb{R}^{|\Omega|}$ given by

$$\mathbf{x}^* = \underset{\mathbf{x} \in \mathbb{R}^R}{\arg\min} \|\mathbf{Px} - \mathbf{q}\|_2. \quad (7)$$

If there exists a tall structured matrix $\mathbf{A} \in \mathbb{R}^{I \times R}$ with a subset of rows $\Omega \subseteq [I]$ such that $\mathbf{A}_{\Omega} = \mathbf{P}$ (permutations of the rows allowed), then we can lift (7) to a higher-dimensional problem while preserving the optimal solution.

**Lemma 3.1.** *Let* $\mathbf{b} \in \mathbb{R}^I$ *be the lifted response such that* $\mathbf{b}_{\Omega} = \mathbf{q}$ *and* $\mathbf{b}_{\overline{\Omega}}$ *is a free variable. If*

$$(\mathbf{x}^*, \mathbf{b}_{\overline{\Omega}}^*) = \underset{\mathbf{x} \in \mathbb{R}^R, \mathbf{b}_{\overline{\Omega}} \in \mathbb{R}^{I-|\Omega|}}{\arg\min} \|\mathbf{A}\mathbf{x} - \mathbf{b}\|_2, \qquad (8)$$

*then* $\mathbf{x}^*$ *also minimizes* (7)*, i.e., the original linear regression problem* $\min_{\mathbf{x} \in \mathbb{R}^R} \|\mathbf{P}\mathbf{x} - \mathbf{q}\|_2$.

**Lemma 3.2.** *Problem 8 is a convex quadratic program.*

*Remark* 3.3. Problem 8 is not a linear regression problem with (structured) design matrix $\mathbf{A}$ because there are $\mathbf{b}_{\overline{\Omega}}$ variables in the response. However, there is enough structure to employ block minimization to alternate between minimizing $\mathbf{x}$ and $\mathbf{b}_{\overline{\Omega}}$.

### 3.2. Iterative Methods for the Lifted Problem

Iterative methods for solving linear systems and regression problems have a long history and have been used to speed up several algorithms in theory and practice. The algorithms we consider use the exact arithmetic model, but all of these methods can be carried out with numbers with $\log{\kappa}/{\varepsilon}$ bits, where $\kappa$ is the condition number of the matrix (see, e.g., Ghadiri et al. (2023b; 2024)). There is a literature on *inexact* Richardson iteration for solving linear systems, but they require the error $\widehat{\varepsilon}$ to be smaller than than $1/\kappa$, which is not achievable with leverage-score sampling (Golub & Overton, 1988; Golub & van der Vorst, 1997).

**Lemma 3.4** (Preconditioned Richardson iteration, (Lee & Vempala, 2024, Lemma 6.1))**.** *Consider the least-squares problem* $\mathbf{x}^* = \arg\min_{\mathbf{x} \in \mathbb{R}^R} \|\mathbf{P}\mathbf{x} - \mathbf{q}\|$. *Let* $\mathbf{M}$ *be a matrix such that* $\mathbf{P}^{\top}\mathbf{P} \preceq \mathbf{M} \preceq \beta \cdot \mathbf{P}^{\top}\mathbf{P}$ *for some* $\beta \geq 1$, *and consider the Richardson iteration:*

$$\mathbf{x}^{(k+1)} = \mathbf{x}^{(k)} - \mathbf{M}^{-1}(\mathbf{P}^{\top}\mathbf{P}\mathbf{x}^{(k)} - \mathbf{P}^{\top}\mathbf{q}).$$

*Then, we have that*

$$\|\mathbf{x}^{(k+1)} - \mathbf{x}^*\|_{\mathbf{M}} \leq \left(1 - \frac{1}{\beta}\right) \|\mathbf{x}^{(k)} - \mathbf{x}^*\|_{\mathbf{M}}.$$

For the rest of this section, let $\widetilde{\mathbf{P}} \in \mathbb{R}^{I \times R}, \widetilde{\mathbf{q}} \in \mathbb{R}^I$ be the zero-masked lifted matrix and vector such that

$$(\widetilde{\mathbf{P}}_{\Omega}, \widetilde{\mathbf{P}}_{\overline{\Omega}}) = (\mathbf{A}_{\Omega}, \mathbf{0}) \quad \text{and} \quad (\widetilde{\mathbf{q}}_{\Omega}, \widetilde{\mathbf{q}}_{\overline{\Omega}}) = (\mathbf{b}_{\Omega}, \mathbf{0}).$$

We now present a key lemma showing that alternating minimization between $\mathbf{x}$ and $\mathbf{b}_{\overline{\Omega}}$ corresponds to preconditioned Richardson iterations on the original least-squares problem. Below, one can easily check that $\mathbf{A}, \widetilde{\mathbf{P}}$, and $\widetilde{\mathbf{q}}$ in our lifted approach satisfy this condition.

**Lemma 3.5.** *Let* $\mathbf{A}, \widetilde{\mathbf{P}} \in \mathbb{R}^{I \times R}, \widetilde{\mathbf{q}} \in \mathbb{R}^I$ *such that* $\widetilde{\mathbf{P}} - \mathbf{A}$ *and* $\begin{bmatrix} \widetilde{\mathbf{P}} & \widetilde{\mathbf{q}} \end{bmatrix}$ *are orthogonal, i.e.,* $(\widetilde{\mathbf{P}} - \mathbf{A})^{\top} \begin{bmatrix} \widetilde{\mathbf{P}} & \widetilde{\mathbf{q}} \end{bmatrix} = \mathbf{0}$. *Then, the iterative method*

$$\widetilde{\mathbf{q}}^{(k)} = \widetilde{\mathbf{q}} + (\mathbf{A} - \widetilde{\mathbf{P}})\mathbf{x}^{(k)},$$

$$\mathbf{x}^{(k+1)} = \underset{\mathbf{x} \in \mathbb{R}^R}{\arg\min} \|\mathbf{A}\mathbf{x} - \widetilde{\mathbf{q}}^{(k)}\|_2^2,$$

*simulates Richardson iterations with preconditioner* $\mathbf{A}^{\top}\mathbf{A}$ *for the regression problem* $\min_{\mathbf{x}} \|\widetilde{\mathbf{P}}\mathbf{x} - \widetilde{\mathbf{q}}\|_2^2$, *i.e.,*

$$\mathbf{x}^{(k+1)} = \mathbf{x}^{(k)} - (\mathbf{A}^{\top}\mathbf{A})^{-1}(\widetilde{\mathbf{P}}^{\top}\widetilde{\mathbf{P}}\mathbf{x}^{(k)} - \widetilde{\mathbf{P}}^{\top}\widetilde{\mathbf{q}}). \quad (9)$$

*Remark* 3.6. In the tensor completion setting, $\mathbf{A} - \widetilde{\mathbf{P}}$ vanishes over $\Omega$, so $\widetilde{\mathbf{q}}^{(k)}$ only updates entries in $\overline{\Omega}$ while maintaining $\mathbf{q}$ on $\Omega$. Thus, computing $\mathbf{x}^{(k+1)}$ corresponds to

$$\mathbf{x}^{(k+1)} = \underset{\mathbf{x} \in \mathbb{R}^R}{\arg\min} \|\mathbf{A}\mathbf{x} - \widetilde{\mathbf{q}}^{(k)}\|_2^2$$
$$= \underset{\mathbf{x} \in \mathbb{R}^R}{\arg\min} \left\{ \|\mathbf{P}\mathbf{x} - \mathbf{q}\|_2^2 + \|\mathbf{A}_{\overline{\Omega}}(\mathbf{x} - \mathbf{x}^{(k)})\|_2^2 \right\}.$$

### 3.3. Approximately Solving the Lifted Problem

We have shown that alternating minimization for the lifted problem (8) is closely connected to preconditioned Richardson iteration and inherits its convergence guarantees. For this to be useful, we need to use fast regression algorithms for the $\mathbf{x}^{(k+1)}$ updates that *exploit the structure* of $\mathbf{A}$, i.e., when solving $\min_{\mathbf{x}} \|\mathbf{A}\mathbf{x} - \widetilde{\mathbf{q}}^{(k)}\|_2$, where $\mathbf{x}^{(k+1)}$ is the vector produced in iteration $k$ of Algorithm 1.

This is where leverage score sampling comes in to play. We exploit the structure of $\mathbf{A}$ to efficiently compute its leverage scores, and then we solve the regression problem efficiently but *approximately*.

Our next result shows how using approximate least-squares solutions *in each step of block minimization* affects the convergence guarantee of our lifted iterative method.

**Theorem 3.7.** *Let* $\mathbf{A}, \widetilde{\mathbf{P}} \in \mathbb{R}^{I \times R}, \widetilde{\mathbf{q}} \in \mathbb{R}^I$, *and* $\beta \geq 1$ *such that* $\widetilde{\mathbf{P}} - \mathbf{A}$ *and* $\begin{bmatrix} \widetilde{\mathbf{P}} & \widetilde{\mathbf{q}} \end{bmatrix}$ *are orthogonal, and*

$$\widetilde{\mathbf{P}}^{\top}\widetilde{\mathbf{P}} \preceq \mathbf{A}^{\top}\mathbf{A} \preceq \beta \cdot \widetilde{\mathbf{P}}^{\top}\widetilde{\mathbf{P}}.$$

*Let* $\varepsilon \in (0,1), \widehat{\varepsilon} \in [0, 1/\beta^2)$ *and* approx-least-squares *be an algorithm that for any* $\widehat{\mathbf{x}} \in \mathbb{R}^R$ *and* $\mathbf{f} = \widetilde{\mathbf{q}} + (\mathbf{A} - \widetilde{\mathbf{P}})\widehat{\mathbf{x}}$, *computes* $\overline{\mathbf{x}} \in \mathbb{R}^R$ *in time* $O(T)$ *such that*

$$\|\mathbf{A}\overline{\mathbf{x}} - \mathbf{f}\|_2^2 \leq (1 + \widehat{\varepsilon}) \min_{\mathbf{x}} \|\mathbf{A}\mathbf{x} - \mathbf{f}\|_2^2.$$

*Then, Algorithm 1 returns an approximate solution* $\widetilde{\mathbf{x}} \in \mathbb{R}^R$, *using* approx-least-squares *as a subroutine, such that*

$$\|\widetilde{\mathbf{P}}\widetilde{\mathbf{x}} - \widetilde{\mathbf{q}}\|_2^2 \leq \left(1 + \frac{2\widehat{\varepsilon}}{(1/\beta - \sqrt{\widehat{\varepsilon}})^2}\right) \min_{\mathbf{x}} \|\widetilde{\mathbf{P}}\mathbf{x} - \widetilde{\mathbf{q}}\|_2^2$$
$$+ \varepsilon \|\widetilde{\mathbf{P}}(\widetilde{\mathbf{P}}^{\top}\widetilde{\mathbf{P}})^{-1}\widetilde{\mathbf{P}}^{\top}\widetilde{\mathbf{q}}\|_2^2,$$

*in* $O\left(\frac{\beta}{1 - \sqrt{\widehat{\varepsilon}}\beta} \cdot T \log{\beta}/{\varepsilon}\right)$ *time.*

---

**Algorithm 1:** approx-mini-als

---

**Data:** $\mathbf{A}, \widetilde{\mathbf{P}} \in \mathbb{R}^{I \times R}, \widetilde{\mathbf{q}} \in \mathbb{R}^I, \beta \geq 1, \varepsilon \in (0,1)$,
$\quad \widehat{\varepsilon} \in [0, 1/\beta^2)$ with $\widetilde{\mathbf{P}}^\top \widetilde{\mathbf{P}} \preceq \mathbf{A}^\top \mathbf{A} \preceq \beta \cdot \widetilde{\mathbf{P}}^\top \widetilde{\mathbf{P}}$

**Result:** $\widetilde{\mathbf{x}} \in \mathbb{R}^R$

1  Initialize $\mathbf{x}^{(0)} = \mathbf{0}$

2  **for** $k = 0, 1, \ldots, \left\lceil \frac{\log(2\beta/\varepsilon)}{2 \left(1/\beta - \sqrt{\widehat{\varepsilon}}\right)} \right\rceil$ **do**

3  $\quad$ Set $\widetilde{\mathbf{q}}^{(k)} \leftarrow \widetilde{\mathbf{q}} + (\mathbf{A} - \widetilde{\mathbf{P}}) \mathbf{x}^{(k)}$ $\qquad$ // implicit

4  $\quad$ Set $\mathbf{x}^{(k+1)}$ to a vector such that
$\quad\quad \|\mathbf{A}\mathbf{x}^{(k+1)} - \widetilde{\mathbf{q}}^{(k)}\|_2^2 \leq (1+\widehat{\varepsilon}) \min_{\mathbf{x}} \|\mathbf{A}\mathbf{x} - \widetilde{\mathbf{q}}^{(k)}\|_2^2$

5  **return** $\mathbf{x}^{(k)}$

---

*Remark* 3.8. To better understand Theorem 3.7, observe that $\widetilde{\mathbf{P}}^\top \widetilde{\mathbf{P}} = \mathbf{P}^\top \mathbf{P}$ is a $\beta$-spectral approximation of $\mathbf{A}^\top \mathbf{A}$, $\varepsilon$ controls the reducible error $\varepsilon \|\widetilde{\mathbf{P}}\mathbf{x}^*\|_2^2$, and $(1+\widehat{\varepsilon})$ is the error in the approximate least-square update for each $\mathbf{x}^{(k)}$.

**Bounding $\beta$.** First, observe that in the case of TD, we have $\mathbf{P} = \mathbf{A}$, so $\beta = 1$. More generally, if $\text{rank}(\mathbf{A}) = s \leq \min\{I, R\}$ and $\mathbf{A} = \mathbf{U}\mathbf{\Sigma}\mathbf{V}^\top$ is a compressed SVD, then $\mathbf{A}$ is said to satisfy the *standard incoherence condition* with parameter $\mu$ (Chen, 2015) if

$$\max_{i \in [I]} \|\mathbf{e}_i^\top \mathbf{U}\|_2 \leq \sqrt{\frac{\mu s}{I}}, \quad \max_{r \in [R]} \|\mathbf{V}^\top \mathbf{e}_r\|_2 \leq \sqrt{\frac{\mu s}{R}}.$$

The $\|\mathbf{e}_i^\top \mathbf{U}\|_2^2$ and $\|\mathbf{V}^\top \mathbf{e}_r\|_2^2$ values are the leverage scores of the rows and columns of $\mathbf{A}$. Applying Cohen et al. (2015, Lemma 4), if each row of $\mathbf{A}$ is observed with probability $p$ such that $p \geq \frac{c\mu s \log s}{I}$ for some absolute constant $c$, then

$$\frac{1}{2}\mathbf{A}^\top \mathbf{A} \preceq \frac{1}{p}\mathbf{A}_\Omega^\top \mathbf{A}_\Omega \preceq \frac{3}{2}\mathbf{A}^\top \mathbf{A},$$

which gives $\beta = 2/p$. Let $\zeta = \max_{i \in [I]} \|\mathbf{a}_i\|_2$, where $\mathbf{a}_i$ is row $i$ of $\mathbf{A}$. Then, the $\alpha\zeta^2$-ridge leverage scores of $\mathbf{A}$ (i.e., $\mathbf{a}_i^\top (\mathbf{A}^\top \mathbf{A} + \alpha\zeta^2 \cdot \mathbf{I}_R)^{-1}\mathbf{a}_i$), for $\alpha \geq 1$, are at most $1/\alpha$. If $p$ is the observation rate, taking $\alpha = \frac{c \log s}{p}$ gives the required incoherence condition. This can be done by introducing an $\ell_2$-regularization term to the TC optimization problem (i.e., solving a ridge regression problem in each ALS step). Note that usually $\alpha$ can be chosen to be much smaller in practice.

## 4. Sampling Methods for Tensor Completion

We are now ready to efficiently solve the *unstructured* least-squares problem (7) induced by ALS for tensor completion, i.e., for $\mathbf{P} \in \mathbb{R}^{|\Omega| \times R}$ and observations $\mathbf{q} \in \mathbb{R}^{|\Omega|}$, find

$$\mathbf{x}^* = \arg\min_{\mathbf{x} \in \mathbb{R}^R} \|\mathbf{P}\mathbf{x} - \mathbf{q}\|_2^2.$$

As in Algorithm 1, we lift this problem to higher dimension to get a structured design matrix $\mathbf{A}$, and use a known fast

algorithm for approximately solving the structured least-squares problem in each step of approximate-mini-ALS. For a given $\widehat{\varepsilon} \in (0, 1/\beta^2)$, the approximate solver computes a solution $\overline{\mathbf{x}} \in \mathbb{R}^R$ in time $O(T_{\widehat{\varepsilon}})$ such that

$$\|\mathbf{A}\overline{\mathbf{x}} - \mathbf{b}\|_2^2 \leq (1+\widehat{\varepsilon}) \min_{\mathbf{x}} \|\mathbf{A}\mathbf{x} - \mathbf{b}\|_2^2.$$

Therefore, for a desired $\varepsilon_1 \in (0,1)$, we set $\widehat{\varepsilon} = \Theta(\varepsilon_1/\beta^2)$ and use a sufficiently small $\varepsilon \leftarrow \varepsilon_2$ in Theorem 3.7. Putting everything together, Algorithm 1 finds an approximate solution $\widetilde{\mathbf{x}} \in \mathbb{R}^R$ in time $O(\beta T_{\varepsilon_1 \beta^{-2}} \log \frac{\beta}{\varepsilon_2})$ that satisfies

$$\|\mathbf{P}\widetilde{\mathbf{x}} - \mathbf{q}\|_2^2 \leq (1+\varepsilon_1) \|\pi_{\mathbf{P}^\perp} \mathbf{q}\|_2^2 + \varepsilon_2 \|\pi_{\mathbf{P}} \mathbf{q}\|_2^2, \quad (10)$$

where $\pi_{\mathbf{P}}$ and $\pi_{\mathbf{P}^\perp}$ are the orthogonal projection matrices into the column space and null space of $\mathbf{P}$, respectively (see Appendix A.1). With this in hand, we are ready to present the running times of our lifted iterative method for TC problems by combining Theorem 3.7 with state-of-the-art tensor decomposition results based on leverage score sampling.

### 4.1. CP Completion

Each ALS update step for CP completion solves a regression problem where the design matrix is the Khatri–Rao product: for $\mathbf{A}^{(k)} \in \mathbb{R}^{I_k \times R}$, $\mathbf{A}^{\neq k} := \bigodot_{n=1, n \neq k}^N \mathbf{A}^{(n)} \in \mathbb{R}^{I_{\neq k} \times R}$, and $\mathbf{Q} = (\mathbf{X}_{(k)}^\top)_\Omega \in \mathbb{R}^{|\Omega| \times I_k}$,

$$\mathbf{A}^{(k)} \leftarrow \arg\min_{\mathbf{A} \in \mathbb{R}^{I_k \times R}} \left\| (\mathbf{A}^{\neq k})_\Omega \mathbf{A}^\top - \mathbf{Q} \right\|_\text{F}.$$

The design matrix $(\mathbf{A}^{\neq k})_\Omega$ does not necessarily have any structure, so a direct method relies on solving the normal equation, which takes $O(R^\omega + R|\Omega|(R + I_k))$ time. Thus, the running time of *one round* of ALS, i.e., updating all $N$ factors, is $O(N(R^\omega + R^2|\Omega|) + R|\Omega| \sum_{n=1}^N I_n)$.

Previous work on CP tensor decomposition (Cheng et al., 2016; Larsen & Kolda, 2022; Bharadwaj et al., 2023) developed fast methods for efficiently computing the leverage scores of a Khatri–Rao product matrix. In particular, Bharadwaj et al. (2023) designed a data structure for computing and maintaining the leverage scores of $\mathbf{A}^{\neq k}$ during ALS updates. This approach requires sampling $\tilde{O}(R/\varepsilon)$ rows of $\mathbf{A}^{\neq k}$. Due to the Khatri–Rao product structure, each row of $\mathbf{A}^{\neq k}$ can be mapped to a sequence of one choice from the rows of each $\mathbf{A}^{(n)}$ for $n \in [N] \backslash \{k\}$. Hence, sampling a row from $\mathbf{A}^{\neq k}$ is equivalent to the following: for each $n \in [N] \backslash \{k\}$, sample a row from $\mathbf{A}^{(n)}$ according to some conditional distribution given sampled rows from $\mathbf{A}^{(1)}, \ldots, \mathbf{A}^{(n-1)}$, and then compute the Hadamard product of $N-1$ sampled rows. Maintaining the full $I_n$-dimensional vector for a conditional probability for each $n$ is costly, so Bharadwaj et al. (2023) developed a binary tree-based data structure to speed up leverage-score sampling for $\mathbf{A}^{\neq k}$.

Applying Bharadwaj et al. (2023, Corollary 3.3), one round of ALS runs in time $\tilde{O}(\varepsilon^{-1} \sum_{n=1}^{N} (I_n R^2 + NR^3))$. Using their CP TD algorithm as the approximate solver in Algorithm 1, and combining its guarantee with Theorem 3.7, we can extend their approach to CP completion.

**Corollary 4.1.** *There is an ALS CP completion algorithm such that (i) after a factor matrix update, each row of $\mathbf{A}^{(n)}$ satisfies* (10), *and (ii) the total running time of one round is*

$$\tilde{O}\left(\frac{\beta^2}{\varepsilon_1} \sum_{n=1}^{N} (I_n R^2 + NR^3) \log \frac{1}{\varepsilon_2}\right).$$

Note that there is *no dependence* on $|\Omega|$ in the running time due to leverage score sampling, i.e., it runs in sublinear time.

## 4.2. Tucker Completion

Fahrbach et al. (2022) designed block-sketching techniques and fast Kronecker product-matrix multiplication algorithms to exploit the ALS structure for Tucker decomposition.

### 4.2.1. CORE TENSOR UPDATE

Recall that for a Tucker decomposition we use the notation $I := \prod_{n \in [N]} I_n$ and $R := \prod_{n \in [N]} R_n$. The ALS core tensor update in the Tucker completion problem is

$$\mathcal{G} \leftarrow \underset{\mathcal{G}' \in \mathbb{R}^{R_1 \times \cdots \times R_N}}{\arg \min} \left\| \left(\bigotimes_{n=1}^{N} \mathbf{A}^{(n)}\right) \text{vec}(\mathcal{G}') - \text{vec}(\mathcal{X})_\Omega \right\|_2.$$

The design matrix above restricted to $\Omega$ is exactly $\mathbf{P}$ in our general setup.

We compare the running times of the direct method and our lifting approach. In the former, we can compute an exact solution to a least-squares problem $\mathbf{x}^* = (\mathbf{P}^\top \mathbf{P})^{-1} \mathbf{P}^\top \mathbf{q}$ in time $O(|\Omega| R^2 + R^\omega)$.

To achieve a fast lifted method, we solve the second step of Algorithm 1 using the leverage score sampling-based core tensor update algorithm in (Fahrbach et al., 2022, Theorem 1.2) with running time

$$\tilde{O}\left(\sum_{n=1}^{N} \left(I_n R_n + \frac{R_n^\omega N^2}{\varepsilon^2}\right) + \frac{R^{2-\theta^*}}{\varepsilon}\right),$$

where $\theta^* > 0$ is an optimizable constant depending on $\{R_n\}_{n \in [N]}$. Using this as the approx-least-squares subroutine in Theorem 3.7, we achieve the following.

**Corollary 4.2.** *There is an algorithm that computes an ALS Tucker completion core tensor update satisfying* (10) *in time*

$$\tilde{O}\left(\left(\mathsf{C} + \frac{\beta^2 R^{2-\theta^*}}{\varepsilon_1}\right)\beta \log \frac{1}{\varepsilon_2}\right), \qquad (11)$$

*where* $\mathsf{C} := \sum_{n=1}^{N}(I_n R_n + \beta^4 R_n^\omega N^2 \varepsilon_1^{-2})$.

### 4.2.2. FACTOR MATRIX UPDATE

The ALS factor matrix update for $\mathbf{A}^{(k)}$ in the Tucker completion problem is

$$\mathbf{A}^{(k)} \leftarrow \underset{\mathbf{A} \in \mathbb{R}^{I_k \times R_k}}{\arg \min} \left\| \left(\left(\bigotimes_{n=1, n \neq k}^{N} \mathbf{A}^{(n)}\right) \mathbf{G}_{(k)}^\top\right)_\Omega \mathbf{A}^\top - \mathbf{Q} \right\|_{\text{F}},$$

where $\mathbf{Q} = (\mathbf{X}_{(k)}^\top)_\Omega \in \mathbb{R}^{|\Omega| \times I_k}$ is a sparse matrix of observations. The running time of a direct method that solves the normal equation is $O(R_k^\omega + R_k |\Omega|(R_{\neq k} + R_k + I_k))$, where $R_{\neq k} = R/R_k$.

The running time of the sampling-based factor-matrix update for $\mathbf{A}^{(k)}$ in Fahrbach et al. (2022, Theorem 1.2) for the full decomposition problem is

$$\tilde{O}\left(\sum_{n=1}^{N} \left(I_n R_n + \frac{R_n^\omega N^2}{\varepsilon^2} + I_k RR_n\right) + \frac{I_k R_{\neq k}^{2-\theta^*}}{\varepsilon}\right).$$

Combining this result with Theorem 3.7, Algorithm 1 has the following running time for a factor matrix update.

**Corollary 4.3.** *There is an algorithm that computes an ALS Tucker completion factor matrix update for $\mathbf{A}^{(k)}$, with each row of $\mathbf{A}^{(k)}$ satisfying* (10), *in time*

$$\tilde{O}\left(\left(\mathsf{C} + \frac{\beta^2 I_k R_{\neq k}^{2-\theta^*}}{\varepsilon_1} + I_k R \sum_{n=1}^{N} R_n\right)\beta \log \frac{1}{\varepsilon_2}\right),$$

*where* $\mathsf{C} = \sum_{n=1}^{N}(I_n R_n + \beta^4 R_n^\omega N^2 \varepsilon_1^{-2})$.

## 4.3. TT Completion

Each ALS step for TT decomposition solves the following least-squares problem with a Kronecker product-type design matrix: for $\mathbf{A}^{\neq k} := \mathbf{A}_{<k} \otimes \mathbf{A}_{>k}^\top \in \mathbb{R}^{I_{\neq k} \times (R_{k-1} R_k)}$ and $\mathbf{Q} = (\mathbf{X}_{(k)}^\top)_\Omega \in \mathbb{R}^{|\Omega| \times I_k}$,

$$\mathcal{A}^{(k)} \leftarrow \underset{\mathcal{B} \in \mathbb{R}^{R_{k-1} \times I_k \times R_k}}{\arg \min} \left\| \left(\mathbf{A}^{\neq k}\right)_\Omega \left(\mathbf{B}_{(2)}\right)^\top - \mathbf{Q} \right\|_{\text{F}}.$$

Solving this directly with the normal equation takes $O(\bar{R}_k^\omega + \bar{R}_k |\Omega|(\bar{R}_k + I_k))$ time for $\bar{R}_k := R_{k-1} R_k$. Thus, the time for one round of ALS is $O(\sum_{n=1}^{N}(\bar{R}_n^\omega + \bar{R}_n |\Omega|(\bar{R}_n + I_n)))$.

In contrast, Bharadwaj et al. (2024, Corollary 4.4) show that one round of approximate TT-core updates, if $R_n = R$ for all $n \in [N-1]$, can run in time $\tilde{O}(R^4 \varepsilon^{-1} \sum_{n=1}^{N}(N + I_n))$. See Appendix C for technical details.

**Corollary 4.4.** *There is an ALS TT completion algorithm such that (i) after a TT-core update, each row fiber of $\mathcal{A}^{(n)}$ satisfies* (10), *and (ii) the total running time of one round is*

$$\tilde{O}\left(\frac{\beta^2 R^4}{\varepsilon_1} \sum_{n=1}^{N}(N + I_n) \log \frac{1}{\varepsilon_2}\right).$$

# 5. Experiments

In this section, we study the empirical performance of our algorithm and compare it to direct and expectation maximization (EM) methods for a coupled matrix problem and CP completion tasks.[1]

## 5.1. Warm-Up: Coupled Matrix Problem

First we consider the *coupled matrix problem*

$$\mathbf{A}\mathbf{X}\mathbf{B}^\top + \mathbf{C}\mathbf{Y}\mathbf{D}^\top = \mathbf{E},$$

where $\mathbf{A}, \mathbf{B}, \mathbf{C}, \mathbf{D} \in \mathbb{R}^{n \times d}$ are given, $\mathbf{X}, \mathbf{Y} \in \mathbb{R}^{d \times d}$ are unknown, and $\mathbf{E} \in \mathbb{R}^{n \times n}$ is a matrix with half of its entries randomly revealed (Baksalary & Kala, 1980). For fixed $\mathbf{Y}$, we compute $\mathbf{X}$ by solving the Kronecker regression problem

$$\arg\min_{\mathbf{X} \in \mathbb{R}^{d \times d}} \|(\mathbf{B} \otimes \mathbf{A})\operatorname{vec}(\mathbf{X}) - \operatorname{vec}(\mathbf{E} - \mathbf{C}\mathbf{Y}\mathbf{D}^\top)\|_2 . \quad (12)$$

The matrix $\mathbf{Y}$ can be updated by solving a similar regression problem. Therefore, we can apply an alternating minimization algorithm to compute $\mathbf{X}$ and $\mathbf{Y}$. For these experiments, we initialize $\mathbf{X} = \mathbf{Y} = \mathbf{I}$. We present the results in Figure 1, which are averaged over five trials.

**Data generation.** The entries of $\mathbf{A}, \mathbf{B}, \mathbf{C}, \mathbf{D}, \mathbf{X}, \mathbf{Y}$ are sampled independently from a uniform distribution on $[0, 1)$, and then we set $\mathbf{E} \leftarrow \mathbf{A}\mathbf{X}\mathbf{B}^\top + \mathbf{C}\mathbf{Y}\mathbf{D}^\top$. We consider the setting where half of the entries of $\mathbf{E}$ are observed (chosen uniformly at random). We set $n = 2000$ and $d = 10$. Note that since we observe a subset of entries of $\mathbf{E}$, the Kronecker regression structure is lost.

**Algorithms.** We compare the `direct`, `mini-als`, and `approximate-mini-als` methods. The latter two use *adaptive* step sizes based on the trajectory of the iterates $\mathbf{x}^{(k)}$ (see Appendix D.1.2 for details). `direct` solves the normal equation in each ALS step and runs in $O(|\Omega|R^2 + R^3)$ time since it computes $(\mathbf{P}^\top\mathbf{P})^{-1}$. `mini-als` is Algorithm 1 with $\widehat{\varepsilon} = 0$ and $\varepsilon > 0$. `mini-als` uses the Kronecker product properties $((\mathbf{B} \otimes \mathbf{A})^\top(\mathbf{B} \otimes \mathbf{A}))^{-1} = (\mathbf{B}^\top\mathbf{B})^{-1} \otimes (\mathbf{A}^\top\mathbf{A})^{-1}$ and $(\mathbf{B} \otimes \mathbf{A})\operatorname{vec}(\mathbf{X}) = \operatorname{vec}(\mathbf{A}\mathbf{X}\mathbf{B}^\top)$ for improved efficiency. `approximate-mini-als` uses leverage score sampling for Kronecker products similar to Fahrbach et al. (2022); Diao et al. (2019), which is a direct application of Corollary 4.2. For leverage score sampling, we sample $1\%$ of rows in each iteration of `approximate-mini-als`.

**Results.** The left plot in Figure 1 shows the total running time of these three algorithms across all ALS iterations. The right plots shows the *mean squared error* (MSE) at each step of ALS. Since `approximate-mini-als` is stochastic, it

[1]The code is available at https://github.com/fahrbach/fast-tensor-completion.

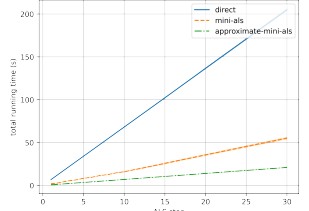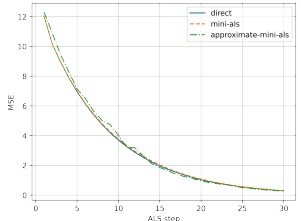

*Figure 1.* Coupled matrix results for $\mathbf{E} \in \mathbb{R}^{n \times n}$, $\mathbf{X}, \mathbf{Y} \in \mathbb{R}^{d \times d}$ with $n = 2000$ and $d = 10$ that compare the direct method, mini-ALS, and approximate-mini-ALS via leverage score sampling.

does not necessarily attain the minimal error in the matrix's kernel space. i.e., the first term on the right-hand side of (10) can exceed the minimum error by a factor of $(1 + \varepsilon)$, which allows it to follow a different convergence path and achieve a lower MSE than the other methods around step 15 of ALS.

## 5.2. CP Completion

Now we compare methods for the CP completion task. For a given tensor $\mathfrak{X}$ and sample ratio $p \in [0, 1]$, let $\mathfrak{X}_\Omega$ be a partially observed tensor with a random $p$ fraction of entries revealed. We fit $\mathfrak{X}_\Omega$ with a rank-$R$ CP decomposition by minimizing the training *relative reconstruction error* (RRE) $\|(\widehat{\mathfrak{X}} - \mathfrak{X})_\Omega\|_\mathrm{F}/\|\mathfrak{X}_\Omega\|_\mathrm{F}$ using different ALS algorithms.

**Datasets.** We consider two real-world tensors. CARDIAC-MRI is a $256 \times 256 \times 14 \times 20$ tensor of MRI measurements indexed by $(x, y, z, t)$ where $(x, y, z)$ is a point in space and $t$ corresponds to time. HYPERSPECTRAL is $1024 \times 1344 \times 33$ tensor of time-lapse hyperspectral radiance images (Nascimento et al., 2016). We also consider synthetic low-rank CP and Tucker tensors in Appendix D.1.1.

**Algorithms.** We compare `direct`, `parafac`, `mini-als`, and `accelerated-mini-als` methods. `direct` solves the normal equation in each ALS step for the original problem of the form (4) and has running time $O(|\Omega|R^2 + R^3)$ since it computes $(\mathbf{A}_\Omega^\top\mathbf{A}_\Omega)^{-1}$. `mini-als` is our lifting approach in Algorithm 1 with $\widehat{\varepsilon} = 0$ and $\varepsilon > 0$. `parafac` is the EM algorithm of Tomasi & Bro (2005), which is equivalent to running Algorithm 1 for exactly one iteration in each step of the ALS algorithm. `accelerated-mini-als` uses adaptive step sizes based on the trajectory of the iterates $\mathbf{x}^{(k)}$.

**Results.** In Figure 2, we present plots for CARDIAC-MRI in the top row and HYPERSPECTRAL in the bottom row.

- In the first column, we set $p = 0.1$ and sweep over the rank $R$ using `direct` to demonstrate how the train RRE (solid line) and test RRE $\|\widehat{\mathfrak{X}} - \mathfrak{X}\|_\mathrm{F}/\|\mathfrak{X}\|_\mathrm{F}$ (dashed line)

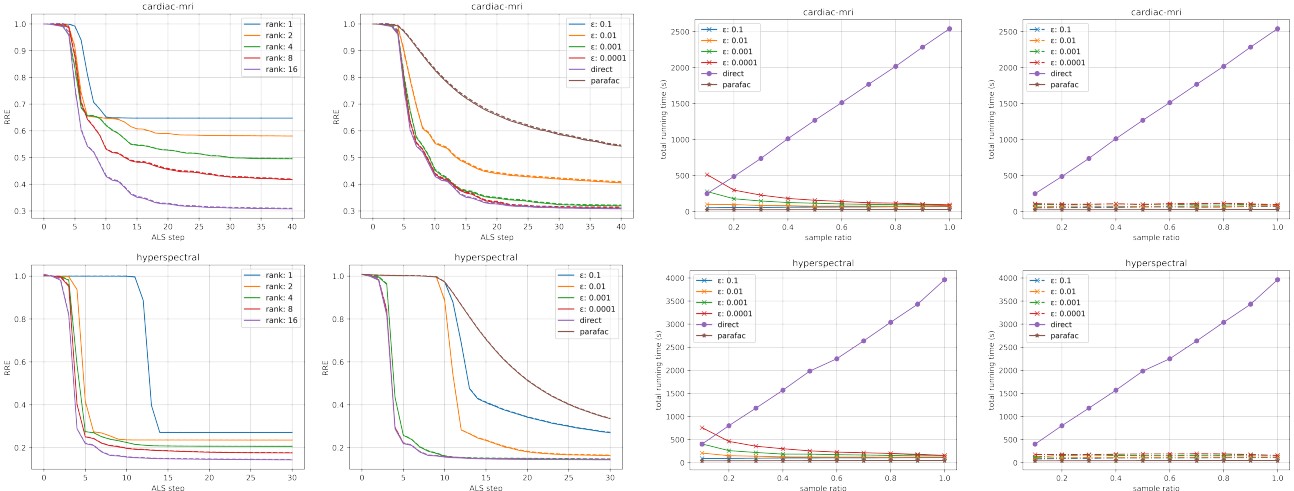

*Figure 2.* Algorithm comparison for a low-rank CP completion task on the CARDIAC-MRI and HYPERSPECTRAL tensor datasets. The first column illustrates convergence rates and relative reconstruction error (RRE) of the ALS algorithm for various CP-decomposition ranks. The second column plots RRE over ALS steps for `direct`, `parafac`, and our method (`mini-als`) under different $\varepsilon$ values. The third column shows total running time of these algorithms at varying sample ratios, i.e., number of observed entries. Finally, the fourth column displays total running times for `direct`, `parafac`, and `accelerated-min-als` (instead of `mini-als`) across different $\varepsilon$ settings.

decrease as a function of $R$ and the ALS step count. Note that the test RRE is the loss on the entire (mostly unobserved) tensor, and is slightly above the train RRE curves at all times.

- In the second column, we fix the parameters $(p, R) = (0.1, 16)$ and study the solution quality of `mini-als` for different values of $\varepsilon$ compared to `direct` and `parafac`. As $\varepsilon$ decreases, we recover solutions in each step of ALS that are as good as `direct`, which agrees with the claim that lifting simulates the preconditioned Richardson iteration (Lemma 3.5).

- In the third column, we sweep over $p$ for $R = 16$ and plot the *total* running time of `direct`, `parafac`, and `mini-als` for 10 rounds of ALS. The running time of `direct` increases linearly in $p \propto |\Omega|$ as expected. Interestingly, the running time of `mini-als` decreases as $|\Omega|$ increases, for $\varepsilon < 0.1$, since the lifted (structured) matrix $\mathbf{A}^\top \mathbf{A}$ becomes a better preconditioner for the TC design matrix $\mathbf{A}_\Omega$ (i.e., $\beta \to 1$ as $p \to 1$). This means `mini-als` needs fewer iterations to converge. Finally, `parafac` performs one mini-ALS iteration in each ALS step and is therefore faster but converges at a slower rate.

- In the fourth column, we compare the total running time of the `accelerated-mini-als` algorithm (dash-dot lines) to `mini-als` (third column). Our accelerated method extrapolates the trajectory of the iterates $\mathbf{x}^{(k)}$ during mini-ALS using a geometric series, which allows us to solve collinear iterates in one step. We

illustrate this idea with Figure 4 in Appendix D.1.2. `accelerated-mini-als` achieves better solution quality than `mini-als` for a given $\varepsilon$ and always runs faster, especially for small values of $p$.

## 6. Conclusion

This work introduces a novel lifting approach for tensor completion. We build on fast sketching-based TD algorithms as ALS subroutines, extending their guarantees to the TC setting and establishing novel connections to iterative methods. We prove guarantees for the convergence rate of an approximate version of the Richardson iteration, and we study how these algorithms perform in practice on real-world tensors. One interesting future direction is to analyze the speedup that adaptive step sizes give in `accelerated-mini-als`.

## Acknowledgements

We would like to thank Arkadi Nemirovski and Devavrat Shah for insightful discussions. MG and AJ acknowledge the support of ARO MURI W911-NF-19-1-0217 and ONR N00014-23-1-2299. YK is supported in part by NSF Award CCF-2106444, and did this work as a student researcher at Google Research.

## Impact Statement

This paper presents work whose goal is to advance the field of machine learning. There are many potential consequences of our work, but none that we feel must be highlighted here.

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

# A. Missing Details for [Section 2](#)

## A.1. Least-Squares Linear Regression

For a design matrix $\mathbf{A} \in \mathbb{R}^{n \times d}$ and response $\mathbf{b} \in \mathbb{R}^n$, consider the least-squares problem

$$\mathbf{x}^* = \arg\min_{\mathbf{x} \in \mathbb{R}^d} \|\mathbf{A}\mathbf{x} - \mathbf{b}\|_2 \,.$$

The solution $\mathbf{x}^*$ can be obtained by solving the *normal equation* $\mathbf{A}^\top \mathbf{A}\mathbf{x}^* = \mathbf{A}^\top \mathbf{b}$. Therefore, $\mathbf{x}^* = (\mathbf{A}^\top \mathbf{A})^{-1} \mathbf{A}^\top \mathbf{b}$. If $\mathbf{A}^\top \mathbf{A}$ is singular, then we can use the *pseudoinverse* $\mathbf{A}^+$.

The *orthogonal projection matrix* $\pi_{\mathbf{A}} : \mathbb{R}^n \to \mathbb{R}^n$ onto the image space of $\mathbf{A}$ is defined by $\pi_{\mathbf{A}} = \mathbf{A} (\mathbf{A}^\top \mathbf{A})^{-1} \mathbf{A}^\top$, and satisfies $\pi_{\mathbf{A}}^2 = \pi_{\mathbf{A}}$ and $\|\pi_{\mathbf{A}} \mathbf{v}\|_2 \leq \|\mathbf{v}\|_2$ for any $\mathbf{v} \in \mathbb{R}^n$. Recall that any $\mathbf{v} \in \mathbb{R}^n$ can be uniquely decomposed as $\mathbf{v} = \pi_{\mathbf{A}} \mathbf{v} + \pi_{\mathbf{A}\perp} \mathbf{v}$, where $\pi_{\mathbf{A}\perp} = \mathbf{I}_n - \pi_{\mathbf{A}}$ is the orthogonal projection to the orthogonal subspace of $\mathrm{colsp}(\mathbf{A})$.

Given $\mathbf{b} = \pi_{\mathbf{A}} \mathbf{b} + \pi_{\mathbf{A}\perp} \mathbf{b}$, the first term is the *reducible error* by regressing $\mathbf{b}$ on $\mathbf{x}$, i.e., taking the optimum $\mathbf{x}^*$ so that $\mathbf{A}\mathbf{x}^* = \pi_{\mathbf{A}} \mathbf{b}$. The second term $\pi_{\mathbf{A}\perp} \mathbf{b}$ is the *irreducible error*, i.e., $\min \|\mathbf{A}\mathbf{x} - \mathbf{b}\|_2 = \|\pi_{\mathbf{A}\perp} \mathbf{b}\|_2$.

## A.2. Leverage Score Sampling for Tensor Decomposition

ALS formulations show how each tensor decomposition step reduces to solving a least-squares problem of the form $\min_{\mathbf{x}} \|\mathbf{A}\mathbf{x} - \mathbf{b}\|_2$ with a highly structured $\mathbf{A}$. While we can find the optimum in closed form via $(\mathbf{A}^\top \mathbf{A})^+ \mathbf{A}^\top \mathbf{b}$, matrix $\mathbf{A}$ has $I = I_1 \cdots I_N$ rows corresponding to each entry of the tensor (i.e., it is a tall skinny matrix), which can make using the normal equation challenging in practice.

Randomized sketching methods are a popular approach to approximately solving this problem with faster running times with high probability. In general, these approach sample rows of $\mathbf{A}$ according to the probability distribution defined by the *leverage scores* of rows, resulting in a random sketching matrix $\mathbf{S}$ whose height is much smaller than that of $\mathbf{A}$. For a matrix $\mathbf{A} \in \mathbb{R}^{I \times R}$ with $(I \gg R)$, the leverage scores of $\mathbf{A}$ is the vector $\ell \in [0,1]^I$ defined by

$$\ell_i \stackrel{\mathrm{def}}{=} \left( \mathbf{A} (\mathbf{A}^\top \mathbf{A})^+ \mathbf{A}^\top \right)_{ii} \,.$$

Then, for a given $\varepsilon, \delta \in (0,1)$, the sketching algorithm samples $\tilde{O}(R/\varepsilon\delta)$ many rows, where the $i$-th row is drawn with probability $\ell_i / \sum_i \ell_i = \ell_i / \mathrm{rank}(\mathbf{A})$. With probability at least $1 - \delta$, we can guarantee that

$$\min_{\mathbf{x}} \|\mathbf{S}\mathbf{A}\mathbf{x} - \mathbf{S}\mathbf{b}\|_2 \leq (1 + \varepsilon) \min_{\mathbf{x}} \|\mathbf{A}\mathbf{x} - \mathbf{b}\|_2 \,.$$

The reduced number of rows in $\mathbf{S}\mathbf{A}$ leads to better running times for the least-squares solves. However, naïvely computing leverage scores takes as long as computing the closed-form optimum since we need to compute $(\mathbf{A}^\top \mathbf{A})^+$. This is where the *structure* of the design matrix $\mathbf{A}$ comes in to play, i.e., to speed up the leverage score computations.

## A.3. Tensor-Train Decomposition

Given a tensor-train (TT) decomposition $\{\mathcal{A}^{(n)}\}_{n=1}^N$ and index $n \in [N]$, define the *left chain* $\mathbf{A}_{<n} \in \mathbb{R}^{(I_1 \cdots I_{n-1}) \times R_{n-1}}$ and the *right chain* $\mathbf{A}_{>n} \in \mathbb{R}^{R_n \times (I_{n+1} \cdots I_N)}$ as:

$$a_{<n}(\underline{i_1 \ldots i_{n-1}}, r_{n-1}) = \sum_{r_0, \ldots, r_{n-1}} \prod_{k=1}^{n-1} a^{(k)}_{r_{k-1} i_k r_k}$$

$$a_{>n}(r_n, \underline{i_{n+1} \ldots i_N}) = \sum_{r_{n+1}, \ldots, r_N} \prod_{k=n+1}^{N} a^{(k)}_{r_{k-1} i_k r_k} \,,$$

where for any $i_s \in [I_s]$ with $s(\neq n) \in [N]$, $\underline{i_1 \ldots i_{n-1}} := 1 + \sum_{k=1}^{n-1} (i_k - 1) \prod_{j=1}^{k-1} I_j$ and $\underline{i_{n+1} \ldots i_N} := 1 + \sum_{k=n+1}^{N} (i_k - 1) \prod_{j=n+1}^{k-1} I_j$. When ALS optimizes $\mathcal{A}^{(n)}$ with all other TT-cores fixed, it solves the regression problem:

$$\mathcal{A}^{(n)} \leftarrow \arg\min_{\mathcal{B} \in \mathbb{R}^{R_{n-1} \times I_n \times R_n}} \left\| (\mathbf{A}_{<n} \otimes \mathbf{A}_{>n}^\top) \mathbf{B}_{(2)}^\top - \mathbf{X}_{(n)}^\top \right\|_{\mathrm{F}} \,,$$

which is equivalent to solving $I_n$ Kronecker regression problems in $\mathbb{R}^{\prod_{k \neq n} I_k}$.

## A.4. Tensor Networks

A *tensor network* (TN) is a powerful framework that can represent any factorization of a tensor, so it can recover the three decompositions above as special cases. A TN decomposition $\mathrm{TN}(\boldsymbol{\mathcal{A}}^{(1)}, \ldots, \boldsymbol{\mathcal{A}}^{(N)})$ represents a given tensor $\boldsymbol{\mathcal{X}}$ with $N$ tensors $\boldsymbol{\mathcal{A}}^{(1)}, \ldots, \boldsymbol{\mathcal{A}}^{(N)}$ and a *tensor diagram*. As in the above decompositions, the goal is to compute

$$\underset{\boldsymbol{\mathcal{A}}^{(1)}, \ldots, \boldsymbol{\mathcal{A}}^{(N)}}{\arg\min} \; \|\boldsymbol{\mathcal{X}} - \mathrm{TN}(\boldsymbol{\mathcal{A}}^{(1)}, \ldots, \boldsymbol{\mathcal{A}}^{(N)})\|_{\mathrm{F}}^2 .$$

A tensor diagram[2] consists of nodes with dangling edges, where a node indicates a tensor, and its dangling edge represents a mode, so that the number of dangling edges is the order of the tensor. For example, a node without an edge indicates a scalar, one with one dangling edge is a vector, and one with two dangling edges is a matrix.

When two dangling edges of two nodes are connected, we say that the two tensors are *contracted* along that mode (i.e., a mode product of those two tensors). For example, when a node with two dangling edges shares one edge with another node with one dangling edge, it indicates a matrix-vector multiplication. Hence, the number of unmatched dangling edges in a tensor diagram corresponds to the order of its representing tensor.

A notable special case is a *fully-connected tensor network* (FCTN) (Zheng et al., 2021) decomposition. It consists of $N$ tensors of the same order, where in its tensor diagram, all pairs of nodes are *connected* as its name suggests.

**ALS for TN decomposition.** Given a TN decomposition $\{\boldsymbol{\mathcal{A}}^{(n)}\}_{n \in [N]}$, when ALS optimizes a tensor $\boldsymbol{\mathcal{A}}^{(n)}$ with all others fixed, it solves a linear regression problem of the form

$$\boldsymbol{\mathcal{A}}^{(n)} \leftarrow \underset{\boldsymbol{\mathcal{B}}}{\arg\min} \, \|\mathbf{A}_{\neq n}\mathbf{B} - \mathbf{X}\|_{\mathrm{F}} ,$$

where $\mathbf{A}_{\neq n}$ is an appropriate matricization depending on $\boldsymbol{\mathcal{A}}^{(1)}, \ldots, \boldsymbol{\mathcal{A}}^{(n-1)}, \boldsymbol{\mathcal{A}}^{(n+1)}, \ldots, \boldsymbol{\mathcal{A}}^{(N)}$, and $\mathbf{B}$ and $\mathbf{X}$ are suitable matricizations of $\boldsymbol{\mathcal{B}}$ and $\boldsymbol{\mathcal{X}}$, respectively. Structure of $\mathbf{A}_{\neq n}$ can be specified through a new tensor diagram obtained by removing the node of $\boldsymbol{\mathcal{A}}^{(n)}$ from the original tensor network diagram.

Just as the ALS approaches for other decomposition algorithms, Malik et al. (2022) proposed a sampling-based approach via leverage scores. First of all, they pointed out that $\mathbf{A}_{\neq n}^{\top}\mathbf{A}_{\neq n}$ can be efficiently computed by exploiting inherent structure of $\mathbf{A}_{\neq n}$ (i.e., contract a series of matched edges in a tensor diagram in an appropriate order). They then presented a leverage-score sampling method that draws rows of $\mathbf{A}_{\neq n}$ without materializing a full probability vector, and in spirit this approach is similar to one used for the CP decomposition in Section 4.1.

**Other tensor decompositions.** We also discuss other important special cases of TN decompositions including the *tensor ring* (TR) (Zhao et al., 2016) and *tensor wheel* (TW) (Wu et al., 2022) decompositions. Both decompositions can be succinctly described through tensor diagram notation.

A TR decomposition consists of $N$ tensors $\boldsymbol{\mathcal{A}}^{(1)}, \ldots, \boldsymbol{\mathcal{A}}^{(N)}$, where for each $k \in [N]$, the node of $\boldsymbol{\mathcal{A}}^{(k)}$ is connected to the two neighboring nodes of $\boldsymbol{\mathcal{A}}^{(k-1)}$ and $\boldsymbol{\mathcal{A}}^{(k+1)}$ (precisely, the superscripts here refer to modulo with respect to $N$) so that the resulting tensor diagram looks exactly like a ring. Since a fast TR decomposition through leverage-score sampling is facilitated by Malik & Becker (2021), our approach of lifting and completion can be straightwardly applied to the TR decomposition.

A TW decomposition consists of one core tensor $\boldsymbol{\mathcal{C}}$ and $N$ factor tensors $\boldsymbol{\mathcal{A}}^{(1)}, \ldots, \boldsymbol{\mathcal{A}}^{(N)}$. As its name suggests, its tensor diagram is essentially that of the tensor ring (consisting of $N$ factor tensors) with the node representing $\boldsymbol{\mathcal{C}}$ connected to all factor tensors. Thanks to a recent work of Wang et al. (2024) using leverage-score sampling for the TW decomposition, our approach can be readily applied as well.

The t-SVD (Kilmer & Martin, 2011) and t-CUR (Tarzanagh & Michailidis, 2018) decompositions for third-order tensors that have been applied to spatiotemporal data. They factor a tensor via the t-product into three tensors; in the case of t-SVD, one of these factors is f-diagonal (see Kilmer & Martin (2011) for definitions). Tarzanagh & Michailidis (2018) propose fast, sampling-based algorithms for t-product–based tensor decompositions. These algorithms can be combined with our lifting method and thus extended to the tensor completion setting.

---

[2]We refer readers to https://tensornetwork.org/diagrams/ for more details.

# B. Missing Analysis for Section 3

## B.1. Proof of Lemma 3.1

**Lemma 3.1.** *Let $\mathbf{b} \in \mathbb{R}^I$ be the lifted response such that $\mathbf{b}_\Omega = \mathbf{q}$ and $\mathbf{b}_{\overline{\Omega}}$ is a free variable. If*

$$(\mathbf{x}^*, \mathbf{b}_{\overline{\Omega}}^*) = \underset{\mathbf{x} \in \mathbb{R}^R, \mathbf{b}_{\overline{\Omega}} \in \mathbb{R}^{I-|\Omega|}}{\arg\min} \|\mathbf{A}\mathbf{x} - \mathbf{b}\|_2 \,, \tag{8}$$

*then $\mathbf{x}^*$ also minimizes (7), i.e., the original linear regression problem $\min_{\mathbf{x} \in \mathbb{R}^R} \|\mathbf{P}\mathbf{x} - \mathbf{q}\|_2$.*

*Proof.* For any $\mathbf{x}$, we have

$$\|\mathbf{A}\mathbf{x} - \mathbf{b}\|_2^2 = \|\mathbf{A}_\Omega \mathbf{x} - \mathbf{b}_\Omega\|_2^2 + \|\mathbf{A}_{\overline{\Omega}} \mathbf{x} - \mathbf{b}_{\overline{\Omega}}\|_2^2 \,,$$

so

$$\min_{\mathbf{x}} \|\mathbf{P}\mathbf{x} - \mathbf{q}\|_2^2 \le \min_{\mathbf{x}, \mathbf{b}_{\overline{\Omega}}} \|\mathbf{A}\mathbf{x} - \mathbf{b}\|_2^2 \,.$$

Moreover, for any $\mathbf{x}$, taking $\mathbf{b}_{\overline{\Omega}} = \mathbf{A}_{\overline{\Omega}} \mathbf{x}$ gives us $\|\mathbf{A}_{\overline{\Omega}} \mathbf{x} - \mathbf{b}_{\overline{\Omega}}\|_2^2 = 0$, which implies that

$$\min_{\mathbf{x}} \|\mathbf{P}\mathbf{x} - \mathbf{q}\|_2^2 \ge \min_{\mathbf{x}, \mathbf{b}_{\overline{\Omega}}} \|\mathbf{A}\mathbf{x} - \mathbf{b}\|_2^2 \,.$$

Therefore,

$$\min_{\mathbf{x}} \|\mathbf{P}\mathbf{x} - \mathbf{q}\|_2^2 = \min_{\mathbf{x}, \mathbf{b}_{\overline{\Omega}}} \|\mathbf{A}\mathbf{x} - \mathbf{b}\|_2^2 \,,$$

and $\mathbf{x}^*$ also minimizes (7). $\square$

## B.2. Proof of Lemma 3.2

**Lemma 3.2.** *Problem 8 is a convex quadratic program.*

*Proof.* Since $\widetilde{\mathbf{q}} \in \mathbb{R}^I$ is defined as $\widetilde{\mathbf{q}}_\Omega = \mathbf{b}_\Omega$ and $\widetilde{\mathbf{q}}_{\overline{\Omega}} = \mathbf{0}$, we can write (8) in the following equivalent manner:

$$(\mathbf{x}^*, \mathbf{b}_{\overline{\Omega}}^*) = \underset{\mathbf{x} \in \mathbb{R}^R, \mathbf{b}_{\overline{\Omega}} \in \mathbb{R}^{I-|\Omega|}}{\arg\min} \left\| \begin{bmatrix} \mathbf{A} & -\mathbf{I}_{:,\overline{\Omega}} \end{bmatrix} \begin{bmatrix} \mathbf{x} \\ \mathbf{b}_{\overline{\Omega}} \end{bmatrix} - \widetilde{\mathbf{q}} \right\|_2^2 \,,$$

where $\mathbf{I}$ is the $I \times I$ identity matrix. $\square$

## B.3. Proof of Lemma 3.5

**Lemma 3.5.** *Let $\mathbf{A}, \widetilde{\mathbf{P}} \in \mathbb{R}^{I \times R}$, $\widetilde{\mathbf{q}} \in \mathbb{R}^I$ such that $\widetilde{\mathbf{P}} - \mathbf{A}$ and $\begin{bmatrix} \widetilde{\mathbf{P}} & \widetilde{\mathbf{q}} \end{bmatrix}$ are orthogonal, i.e., $(\widetilde{\mathbf{P}} - \mathbf{A})^\top \begin{bmatrix} \widetilde{\mathbf{P}} & \widetilde{\mathbf{q}} \end{bmatrix} = \mathbf{0}$. Then, the iterative method*

$$\widetilde{\mathbf{q}}^{(k)} = \widetilde{\mathbf{q}} + (\mathbf{A} - \widetilde{\mathbf{P}}) \mathbf{x}^{(k)} \,,$$
$$\mathbf{x}^{(k+1)} = \underset{\mathbf{x} \in \mathbb{R}^R}{\arg\min} \|\mathbf{A}\mathbf{x} - \widetilde{\mathbf{q}}^{(k)}\|_2^2 \,,$$

*simulates Richardson iterations with preconditioner $\mathbf{A}^\top \mathbf{A}$ for the regression problem $\min_{\mathbf{x}} \|\widetilde{\mathbf{P}}\mathbf{x} - \widetilde{\mathbf{q}}\|_2^2$, i.e.,*

$$\mathbf{x}^{(k+1)} = \mathbf{x}^{(k)} - (\mathbf{A}^\top \mathbf{A})^{-1}(\widetilde{\mathbf{P}}^\top \widetilde{\mathbf{P}} \mathbf{x}^{(k)} - \widetilde{\mathbf{P}}^\top \widetilde{\mathbf{q}}) \,. \tag{9}$$

*Proof.* Assume that $\mathbf{A}^\top \mathbf{A}$ is full rank. Solving the normal equation for $\mathbf{x}^{(k+1)}$,

$$\begin{aligned} \mathbf{x}^{(k+1)} &= (\mathbf{A}^\top \mathbf{A})^{-1} \mathbf{A}^\top \widetilde{\mathbf{q}}^{(k)} \\ &= (\mathbf{A}^\top \mathbf{A})^{-1} \mathbf{A}^\top (\widetilde{\mathbf{q}} + (\mathbf{A} - \widetilde{\mathbf{P}}) \mathbf{x}^{(k)}) \\ &= \mathbf{x}^{(k)} - (\mathbf{A}^\top \mathbf{A})^{-1} \mathbf{A}^\top (\widetilde{\mathbf{P}} \mathbf{x}^{(k)} - \widetilde{\mathbf{q}}) \,. \end{aligned}$$

Since $\widetilde{\mathbf{P}} - \mathbf{A}$ and $\begin{bmatrix} \widetilde{\mathbf{P}} & \widetilde{\mathbf{q}} \end{bmatrix}$ are orthogonal,

$$\mathbf{A}^\top(\widetilde{\mathbf{P}}\mathbf{x}^{(k)} - \widetilde{\mathbf{q}}) = \left(\widetilde{\mathbf{P}} - (\widetilde{\mathbf{P}} - \mathbf{A})\right)^\top \begin{bmatrix} \widetilde{\mathbf{P}} & \widetilde{\mathbf{q}} \end{bmatrix} \begin{bmatrix} \mathbf{x}^{(k)} \\ -1 \end{bmatrix}$$
$$= \widetilde{\mathbf{P}}^\top(\widetilde{\mathbf{P}}\mathbf{x}^{(k)} - \widetilde{\mathbf{q}}).$$

Therefore,

$$\mathbf{x}^{(k+1)} = \mathbf{x}^{(k)} - (\mathbf{A}^\top\mathbf{A})^{-1}(\widetilde{\mathbf{P}}^\top\widetilde{\mathbf{P}}\mathbf{x}^{(k)} - \widetilde{\mathbf{P}}^\top\widetilde{\mathbf{q}}),$$

which completes the proof. $\qquad\square$

### B.4. Proof of Theorem 3.7

**Theorem 3.7.** *Let* $\mathbf{A}, \widetilde{\mathbf{P}} \in \mathbb{R}^{I \times R}, \widetilde{\mathbf{q}} \in \mathbb{R}^I$, *and* $\beta \geq 1$ *such that* $\widetilde{\mathbf{P}} - \mathbf{A}$ *and* $\begin{bmatrix} \widetilde{\mathbf{P}} & \widetilde{\mathbf{q}} \end{bmatrix}$ *are orthogonal, and*

$$\widetilde{\mathbf{P}}^\top\widetilde{\mathbf{P}} \preceq \mathbf{A}^\top\mathbf{A} \preceq \beta \cdot \widetilde{\mathbf{P}}^\top\widetilde{\mathbf{P}}.$$

*Let* $\varepsilon \in (0, 1), \widehat{\varepsilon} \in [0, 1/\beta^2)$ *and* approx-least-squares *be an algorithm that for any* $\widehat{\mathbf{x}} \in \mathbb{R}^R$ *and* $\mathbf{f} = \widetilde{\mathbf{q}} + (\mathbf{A} - \widetilde{\mathbf{P}})\widehat{\mathbf{x}}$, *computes* $\overline{\mathbf{x}} \in \mathbb{R}^R$ *in time* $O(T)$ *such that*

$$\|\mathbf{A}\overline{\mathbf{x}} - \mathbf{f}\|_2^2 \leq (1 + \widehat{\varepsilon}) \min_{\mathbf{x}} \|\mathbf{A}\mathbf{x} - \mathbf{f}\|_2^2.$$

*Then, Algorithm 1 returns an approximate solution* $\widetilde{\mathbf{x}} \in \mathbb{R}^R$, *using* approx-least-squares *as a subroutine, such that*

$$\|\widetilde{\mathbf{P}}\widetilde{\mathbf{x}} - \widetilde{\mathbf{q}}\|_2^2 \leq \left(1 + \frac{2\widehat{\varepsilon}}{(1/\beta - \sqrt{\widehat{\varepsilon}})^2}\right) \min_{\mathbf{x}} \|\widetilde{\mathbf{P}}\mathbf{x} - \widetilde{\mathbf{q}}\|_2^2$$
$$+ \varepsilon \|\widetilde{\mathbf{P}}(\widetilde{\mathbf{P}}^\top\widetilde{\mathbf{P}})^{-1}\widetilde{\mathbf{P}}^\top\widetilde{\mathbf{q}}\|_2^2,$$

*in* $O\left(\frac{\beta}{1 - \sqrt{\widehat{\varepsilon}}\beta} \cdot T \log \beta/\varepsilon\right)$ *time.*

*Proof.* We show that Algorithm 1 gives the desired output. Suppose approx-least-squares yields $\mathbf{x}^{(k+1)}$ for given inputs $\mathbf{A}, \widetilde{\mathbf{P}}, \widetilde{\mathbf{q}}$, and $\widetilde{\mathbf{q}}^{(k)}$ (i.e., $\widehat{\mathbf{x}} \leftarrow \mathbf{x}^{(k)}, \mathbf{f} \leftarrow \widetilde{\mathbf{q}}^{(k)}$, and $\overline{\mathbf{x}} \leftarrow \mathbf{x}^{(k+1)}$), which satisfies

$$\|\mathbf{A}\mathbf{x}^{(k+1)} - \widetilde{\mathbf{q}}^{(k)}\|_2^2 \leq (1 + \widehat{\varepsilon}) \min_{\mathbf{x}} \|\mathbf{A}\mathbf{x} - \widetilde{\mathbf{q}}^{(k)}\|_2^2 = (1 + \widehat{\varepsilon}) \|\pi_{\mathbf{A}^\perp}\widetilde{\mathbf{q}}^{(k)}\|_2^2.$$

We can also decompose the LHS using $\widetilde{\mathbf{q}}^{(k)} = \pi_{\mathbf{A}}\widetilde{\mathbf{q}}^{(k)} + \pi_{\mathbf{A}^\perp}\widetilde{\mathbf{q}}^{(k)}$ as follows:

$$\|\mathbf{A}\mathbf{x}^{(k+1)} - \widetilde{\mathbf{q}}^{(k)}\|_2^2 = \|\mathbf{A}\mathbf{x}^{(k+1)} - \pi_{\mathbf{A}}\widetilde{\mathbf{q}}^{(k)}\|_2^2 + \|\pi_{\mathbf{A}^\perp}\widetilde{\mathbf{q}}^{(k)}\|_2^2.$$

Combining the above, we get
$$\|\mathbf{A}\mathbf{x}^{(k+1)} - \pi_{\mathbf{A}}\widetilde{\mathbf{q}}^{(k)}\|_2^2 \leq \widehat{\varepsilon} \|\pi_{\mathbf{A}^\perp}\widetilde{\mathbf{q}}^{(k)}\|_2^2.$$

Denoting $\mathbf{x}^* = (\widetilde{\mathbf{P}}^\top\widetilde{\mathbf{P}})^{-1}\widetilde{\mathbf{P}}^\top\widetilde{\mathbf{q}} = \arg\min_{\mathbf{x}} \|\mathbf{B}\mathbf{x} - \widetilde{\mathbf{q}}\|_2$ and using the triangle inequality,

$$\|\mathbf{A}\mathbf{x}^{(k+1)} - \mathbf{A}\mathbf{x}^*\|_2 \leq \|\mathbf{A}\mathbf{x}^{(k+1)} - \pi_{\mathbf{A}}\widetilde{\mathbf{q}}^{(k)}\|_2 + \|\pi_{\mathbf{A}}\widetilde{\mathbf{q}}^{(k)} - \mathbf{A}\mathbf{x}^*\|_2$$
$$\leq \sqrt{\widehat{\varepsilon}} \|\pi_{\mathbf{A}^\perp}\widetilde{\mathbf{q}}^{(k)}\|_2 + \|\pi_{\mathbf{A}}\widetilde{\mathbf{q}}^{(k)} - \mathbf{A}\mathbf{x}^*\|_2. \tag{13}$$

We now bound each term in the RHS. As for the second term, since $\widetilde{\mathbf{q}}^{(k)} = \widetilde{\mathbf{q}} + (\mathbf{A} - \widetilde{\mathbf{P}})\mathbf{x}^{(k)}$ and $(\widetilde{\mathbf{P}} - \mathbf{A})^\top \begin{bmatrix} \widetilde{\mathbf{P}} & \widetilde{\mathbf{q}} \end{bmatrix} = 0$, by Lemma 3.5,

$$(\mathbf{A}^\top\mathbf{A})^{-1}\mathbf{A}^\top\widetilde{\mathbf{q}}^{(k)} = \arg\min_{\mathbf{x}} \|\mathbf{A}\mathbf{x} - \widetilde{\mathbf{q}}^{(k)}\|_2$$
$$= \mathbf{x}^{(k)} - (\mathbf{A}^\top\mathbf{A})^{-1}(\widetilde{\mathbf{P}}^\top\widetilde{\mathbf{P}}\mathbf{x}^{(k)} - \widetilde{\mathbf{P}}^\top\widetilde{\mathbf{q}}),$$

which is exactly a Richardson iteration with preconditioner $\mathbf{M} \leftarrow \mathbf{A}^\top \mathbf{A}$ in Lemma 3.4 (satisfying $\widetilde{\mathbf{P}}^\top \widetilde{\mathbf{P}} \preceq \mathbf{A}^\top \mathbf{A} \preceq \beta \widetilde{\mathbf{P}}^\top \widetilde{\mathbf{P}}$). Thus, $\|(\mathbf{A}^\top \mathbf{A})^{-1} \mathbf{A}^\top \widetilde{\mathbf{q}}^{(k)} - \mathbf{x}^*\|_{\mathbf{A}^\top \mathbf{A}} \leq (1 - \beta^{-1}) \|\mathbf{x}^{(k)} - \mathbf{x}^*\|_{\mathbf{A}^\top \mathbf{A}}$, and

$$\|\pi_{\mathbf{A}} \widetilde{\mathbf{q}}^{(k)} - \mathbf{A}\mathbf{x}^*\|_2 \leq \left(1 - \frac{1}{\beta}\right) \|\mathbf{A}\mathbf{x}^{(k)} - \mathbf{A}\mathbf{x}^*\|_2. \tag{14}$$

Regarding the first term in (13), since $\mathbf{A}\mathbf{x}^{(k)}$ is in the column space of $\mathbf{A}$,

$$\pi_{\mathbf{A}^\perp} \widetilde{\mathbf{q}}^{(k)} = \pi_{\mathbf{A}^\perp} \left(\widetilde{\mathbf{q}} + (\mathbf{A} - \mathbf{B})\mathbf{x}^{(k)}\right) = \pi_{\mathbf{A}^\perp} \left(\widetilde{\mathbf{q}} - \widetilde{\mathbf{P}}\mathbf{x}^{(k)}\right).$$

Therefore,

$$\begin{aligned} \|\pi_{\mathbf{A}^\perp} \widetilde{\mathbf{q}}^{(k)}\|_2^2 &\leq \|\widetilde{\mathbf{q}} - \widetilde{\mathbf{P}}\mathbf{x}^{(k)}\|_2^2 \\ &= \|\widetilde{\mathbf{P}}\mathbf{x}^* - \widetilde{\mathbf{P}}\mathbf{x}^{(k)}\|_2^2 + \min_{\mathbf{x}} \|\widetilde{\mathbf{P}}\mathbf{x} - \widetilde{\mathbf{q}}\|_2^2 \\ &\leq \|\mathbf{A}\mathbf{x}^* - \mathbf{A}\mathbf{x}^{(k)}\|_2^2 + \min_{\mathbf{x}} \|\widetilde{\mathbf{P}}\mathbf{x} - \widetilde{\mathbf{q}}\|_2^2, \end{aligned}$$

where the last inequality follows from $\widetilde{\mathbf{P}}^\top \widetilde{\mathbf{P}} \preceq \mathbf{A}^\top \mathbf{A}$. Thus,

$$\|\pi_{\mathbf{A}^\perp} \widetilde{\mathbf{q}}^{(k)}\|_2 \leq \|\mathbf{A}\mathbf{x}^* - \mathbf{A}\mathbf{x}^{(k)}\|_2 + \min_{\mathbf{x}} \|\widetilde{\mathbf{P}}\mathbf{x} - \widetilde{\mathbf{q}}\|_2. \tag{15}$$

Combining (13), (14), and (15), we have

$$\|\mathbf{A}\mathbf{x}^{(k+1)} - \mathbf{A}\mathbf{x}^*\|_2 \leq \left(1 - \frac{1}{\beta} + \sqrt{\widehat{\varepsilon}}\right) \|\mathbf{A}\mathbf{x}^* - \mathbf{A}\mathbf{x}^{(k)}\|_2 + \sqrt{\widehat{\varepsilon}} \min_{\mathbf{x}} \|\widetilde{\mathbf{P}}\mathbf{x} - \widetilde{\mathbf{q}}\|_2.$$

Denoting $\alpha = 1 - \frac{1}{\beta} + \sqrt{\widehat{\varepsilon}}$, by induction, we have

$$\begin{aligned} \|\mathbf{A}\mathbf{x}^{(k)} - \mathbf{A}\mathbf{x}^*\|_2 &\leq \alpha^k \|\mathbf{A}\mathbf{x}^* - \mathbf{A}\mathbf{x}^{(0)}\|_2 + (1 + \alpha + \alpha^2 + \cdots + \alpha^{k-1}) \times \sqrt{\widehat{\varepsilon}} \min_{\mathbf{x}} \|\widetilde{\mathbf{P}}\mathbf{x} - \widetilde{\mathbf{q}}\|_2 \\ &= \alpha^k \|\mathbf{A}\mathbf{x}^* - \mathbf{A}\mathbf{x}^{(0)}\|_2 + \frac{1 - \alpha^k}{1 - \alpha} \times \sqrt{\widehat{\varepsilon}} \min_{\mathbf{x}} \|\widetilde{\mathbf{P}}\mathbf{x} - \widetilde{\mathbf{q}}\|_2. \end{aligned} \tag{16}$$

We also have

$$\begin{aligned} \|\widetilde{\mathbf{P}}\mathbf{x}^{(k)} - \widetilde{\mathbf{q}}\|_2^2 &= \|\widetilde{\mathbf{P}}\mathbf{x}^{(k)} - \pi_{\widetilde{\mathbf{P}}} \widetilde{\mathbf{q}}\|_2^2 + \|\pi_{\widetilde{\mathbf{P}}^\perp} \widetilde{\mathbf{q}}\|_2^2 \\ &= \|\widetilde{\mathbf{P}}\mathbf{x}^{(k)} - \widetilde{\mathbf{P}}\mathbf{x}^*\|_2^2 + \min_{\mathbf{x}} \|\widetilde{\mathbf{P}}\mathbf{x} - \widetilde{\mathbf{q}}\|_2^2. \end{aligned} \tag{17}$$

We then bound the first term by using $\widetilde{\mathbf{P}}^\top \widetilde{\mathbf{P}} \preceq \mathbf{A}^\top \mathbf{A} \preceq \beta \widetilde{\mathbf{P}}^\top \widetilde{\mathbf{P}}$ and (16) as follows:

$$\begin{aligned} \|\widetilde{\mathbf{P}}\mathbf{x}^{(k)} - \widetilde{\mathbf{P}}\mathbf{x}^*\|_2^2 &\leq \|\mathbf{A}\mathbf{x}^{(k)} - \mathbf{A}\mathbf{x}^*\|_2^2 \\ &\leq 2\alpha^{2k} \|\mathbf{A}\mathbf{x}^* - \mathbf{A}\mathbf{x}^{(0)}\|_2^2 + 2\left(\frac{1 - \alpha^k}{1 - \alpha}\right)^2 \times \widehat{\varepsilon} \min_{\mathbf{x}} \|\widetilde{\mathbf{P}}\mathbf{x} - \widetilde{\mathbf{q}}\|_2^2 \\ &\leq 2\beta\alpha^{2k} \|\widetilde{\mathbf{P}}\mathbf{x}^* - \widetilde{\mathbf{P}}\mathbf{x}^{(0)}\|_2^2 + 2\left(\frac{1 - \alpha^k}{1 - \alpha}\right)^2 \times \widehat{\varepsilon} \min_{\mathbf{x}} \|\widetilde{\mathbf{P}}\mathbf{x} - \widetilde{\mathbf{q}}\|_2^2. \end{aligned}$$

Putting this bound back into (17),

$$\|\widetilde{\mathbf{P}}\mathbf{x}^{(k)} - \widetilde{\mathbf{q}}\|_2^2 \leq 2\beta\alpha^{2k} \|\widetilde{\mathbf{P}}\mathbf{x}^* - \widetilde{\mathbf{P}}\mathbf{x}^{(0)}\|_2^2 + \left(1 + 2\widehat{\varepsilon}\left(\frac{1 - \alpha^k}{1 - \alpha}\right)^2\right) \min_{\mathbf{x}} \|\widetilde{\mathbf{P}}\mathbf{x} - \widetilde{\mathbf{q}}\|_2^2.$$

Setting

$$k = \left\lceil \frac{\log(2\beta/\varepsilon)}{2\left(1/\beta - \sqrt{\widehat{\varepsilon}}\right)} \right\rceil,$$

we have

$$\|\widetilde{\mathbf{P}}\mathbf{x}^{(k)} - \widetilde{\mathbf{q}}\|_2^2 \leq \varepsilon \|\widetilde{\mathbf{P}}\mathbf{x}^* - \widetilde{\mathbf{P}}\mathbf{x}^{(0)}\|_2^2 + \left(1 + \frac{2\widehat{\varepsilon}}{(1/\beta - \sqrt{\widehat{\varepsilon}})^2}\right) \min_{\mathbf{x}} \|\widetilde{\mathbf{P}}\mathbf{x} - \widetilde{\mathbf{q}}\|_2^2,$$

which completes the proof with $\mathbf{x}^{(0)} = \mathbf{0}$. $\qquad\square$

## C. Additional Details for Section 4

Bharadwaj et al. (2024) proposed a sampling-based ALS algorithm that relies a *canonical form* of the TT decomposition with respect to the index $k$. Any TT decomposition can be converted to this form through a QR decomposition, which ensures that $(\mathbf{A}^{\neq k})^\top \mathbf{A}^{\neq k} = I_{R_{k-1} R_k}$. It follows that the leverage scores of $\mathbf{A}^{\neq k}$ are simply the diagonal entries of

$$\mathbf{A}^{\neq k}(\mathbf{A}^{\neq k})^\top = (\mathbf{A}_{<k}\mathbf{A}_{<k}^\top) \otimes (\mathbf{A}_{>k}^\top \mathbf{A}_{>k}).$$

It follows from properties of the Kronecker product that

$$\ell_{i^{\neq k}}(\mathbf{A}^{\neq k}) = \ell_{i_{<k}}(\mathbf{A}_{<k}) \cdot \ell_{i_{>k}}(\mathbf{A}_{>k}^\top),$$

where $i^{\neq k} = i_1 \cdots i_{k-1} i_{k+1} \cdots i_N$, $i_{<k} = i_1 \cdots i_{k-1}$, and $i_{>k} = i_{k+1} \cdots i_N$ (see Appendix A.3 for definition). Therefore, efficient leverage score sampling for $\mathbf{A}^{\neq k}$ reduces to that for $\mathbf{A}_{<k}$ and $\mathbf{A}_{>k}$. To this end, Bharadwaj et al. (2024) adopt an approach similar to Bharadwaj et al. (2023) for leverage score-based CP decomposition. Each row of $\mathbf{A}_{<k}$ corresponds to a series of one slice for each third-order tensor $\mathbf{A}^{(n)}$ for $n < k$, which results in a series of conditional sampling steps using a data structure adapted from the one used for CP decomposition.

## D. Additional Details for Section 5

All experiments are implemented with NumPy (Harris et al., 2020) and Tensorly (Kossaifi et al., 2019) on an Apple M2 chip with 8 GB of RAM.

### D.1. CP Completion

#### D.1.1. SYNTHETIC TENSORS

We run the same set of experiments as in Section 5 on two different random low-rank tensors:

- RANDOM-CP is a $100 \times 100 \times 100$ tensor formed by reconstructing a random rank-16 CP decomposition.

- RANDOM-TUCKER is a $100 \times 100 \times 100$ tensor formed by reconstructing a random rank-$(4, 4, 4)$ Tucker decomposition.

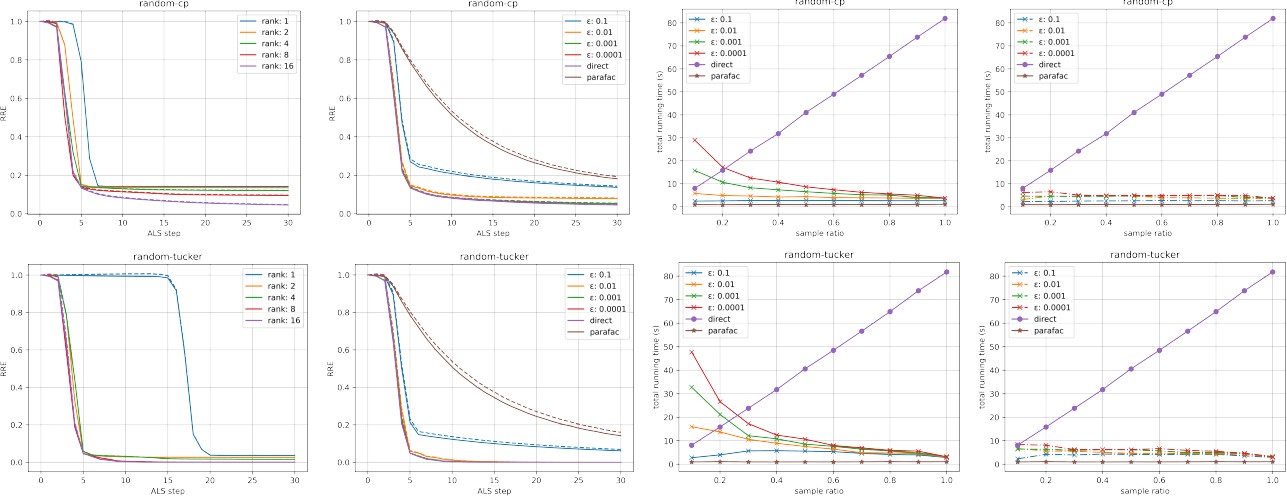

*Figure 3.* Algorithm comparison for a low-rank CP completion task on the RANDOM-CP and RANDOM-TUCKER tensor datasets.

#### D.1.2. ACCELERATED METHODS

We explain how to accelerate the Richardson iteration.

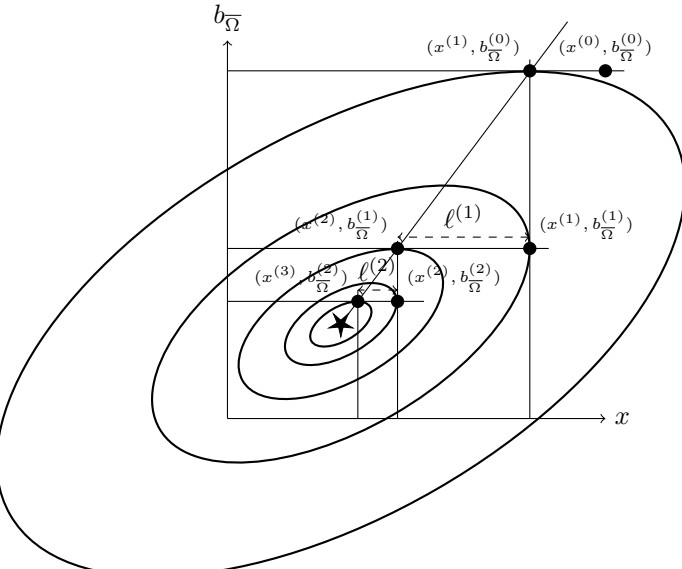

*Figure 4.* In the one-variable case, our approach proceeds in the following order: $(x^{(0)}, b_{\overline{\Omega}}^{(0)}) \to (x^{(1)}, b_{\overline{\Omega}}^{(0)}) \to (x^{(1)}, b_{\overline{\Omega}}^{(1)}) \to \cdots$.

1. For odd iterations (e.g., the first iteration), run mini-ALS normally.

2. For even iterations, compute $\widehat{\mathbf{x}}^{(k+1)}$ using normal mini-ALS, but then set

$$\mathbf{x}^{(k+1)} = \mathbf{x}^{(k)} + \frac{1}{1-\alpha}\left(\widehat{\mathbf{x}}^{(k+1)} - \mathbf{x}^{(k)}\right),$$

where $\alpha = \frac{\|\widehat{\mathbf{x}}^{(k+1)} - \mathbf{x}^{(k)}\|_2}{\|\mathbf{x}^{(k)} - \mathbf{x}^{(k-1)}\|_2}$.

Note that setting $\alpha = 0$ is equivalent to running mini-ALS normally. We explain the intuition behind this accelration with the example in Figure 4, which illustrates the case where $x$ and $b_{\overline{\Omega}}$ both have only one variable. As mentioned previously, the lifted problem is a convex quadratic problem (see Lemma 3.2), and the iterations of mini-ALS alternate between optimizing $x$ and optimizing $b_{\overline{\Omega}}$. In Figure 4, the star ($\star$) denotes the optimal point, and the ellipses denote the level sets of the quadratic function. In the steps where we optimize $x$, we search for the point on the line that is parallel to the $x$ axis that crosses our current point and touches the smallest ellipse among the points on the line. Similarly, when we update $b_{\overline{\Omega}}$ we search on the line that is parallel to the $b_{\overline{\Omega}}$ axis.

Following the points in Figure 4, one can see that the points obtained through our iterations in the one-variable case form similar triangles, where the ratio of corresponding sides for every two consecutive triangles are the same. Therefore, if we denote the side length of the first triangle with 1, then the side length for the next triangles are $\alpha, \alpha^2, \alpha^3, \ldots$. Using the notation in Figure 4, $\alpha = \frac{\ell^{(2)}}{\ell^{(1)}}$. The sum over this geometric series is equal to $\frac{1}{1-\alpha}$, which inspires our *adaptive step size* in even iterations.

Note that in the one-variable case, we can recover the optimal solution with only two iterations as the lines connecting $(x^{(k)}, b_{\overline{\Omega}}^{(k-1)})$ go through the optimal solution. While this does not necessarily hold in higher-dimensional settings, our experiments demonstrate that acceleration improves the number of iterations and the running time of our approach significantly, especially when the number of observed entries are small (i.e., when $\beta$ is large).

