# OpenReview forum: "Fast Tensor Completion via Approximate Richardson Iteration"
_ICML.cc/2025/Conference — ICML 2025 poster_

### Official Review · Reviewer_NbQN · 2025-02-26

**Overall Recommendation:** 4

**Summary:**

This paper addresses the computational challenges of Tensor Completion (TC) by proposing a novel method that integrates Tensor Decompositions (TDs)—including CANDECOMP/PARAFAC (CP), Tucker, and Tensor Train (TT)—with a lifting framework.

The authors reformulate TC as a structured tensor decomposition problem, effectively reducing its computational complexity. This reformulation enables the application of efficient Alternating Least Squares (ALS) algorithms. By interpreting the algorithm as a preconditioned Richardson iteration, the authors establish rigorous theoretical convergence guarantees.

To further enhance efficiency, the method incorporates sketching techniques via leverage score sampling, strategically subsampling tensor fibers to accelerate computations.

Comprehensive experiments on synthetic and real-world datasets demonstrate that the proposed approach achieves significant speedups over state-of-the-art TC methods while maintaining competitive accuracy. The integration of lifting with sketching and convergence analysis presents a key advancement in scalable tensor completion.

## update after rebuttal
I thank very much the authors for clarifying the issues I pointed out and their promise to improve the final version accordingly. I keep my original score.

**Claims And Evidence:**

The claims in the submission are generally well-supported by theoretical analysis and empirical results. The main claims are:

-	Reduction in Computational Complexity: The paper provides theoretical justifications for the efficiency of the proposed method using leverage score sampling and the lifting approach. Running time analyses for CP, Tucker, and TT decompositions are given. Empirical results show significant speed improvements over standard ALS-based CP TC methods.

-	Interpretation as a Preconditioned Richardson Iteration: Theoretical derivations (Lemma 3.5) show that the mini-ALS algorithm simulates Richardson iteration.

**Essential References Not Discussed:**

No, I think the papers included in the references are a good selection of relevant works in this field.

**Experimental Designs Or Analyses:**

The experimental designs and analysis are correct and useful. However, further clarification is needed (see detailed comments below).

**Methods And Evaluation Criteria:**

The proposed methods and evaluation criteria are reasonable and relevant for the tensor completion (TC) problem. However, certain aspects require further clarification (see detailed comments below).

**Other Comments Or Suggestions:**

Here, I provide my detailed comments/suggestions to improve clarity and correctness of the paper:

-	The general TC problem of eq. (1) is not alligned with the approach of the current paper where a TD approach is used meaning that the approximation of the data tensor is a low-rank tensor decomposition (CP, Tucker or TT) where its factors are the minimizing parameter \theta. In other words, the regularization term of eq. (1) is not needed nor used in this paper.

-	The statement “Rank constraints can be incorporated into (1) by including appropriate penalty terms in R(\theta)” is confusing since in this paper, the low rank constraint is explicitly imposed by chosing the rank of the decomposition.

-	Experimental results requires further clarification:

1)	Figure 1 show that the approximate-mini-als algorithm achieves lower error than the other algorithms for a range of steps around step 15. This is surprising and should be explained.

2)	In the paper, it is not clearly explained what the authors refers as direct and parafac methods.

3)	Figure 2 is confusing, for example there is no explanation about what do the plots in 3rd and 4th columns refer to. The caption should be improved.

**Other Strengths And Weaknesses:**

Strengths:

-	Theoretical Contribution: A useful connection between TC and preconditioned Richardson iteration, providing rigorous convergence guarantees (Theorem 3.7) is provided. The lifting approach restores structure in TC regression problems, enabling the use of fast TD algorithms as subroutines. This bridges iterative optimization and tensor algebra in a principled manner.

-	Generality: The framework applies to multiple tensor decompositions (CP, Tucker, TT), demonstrating flexibility. By using TD algorithms as black-box subroutines, the method benefits from future advancements in TD research.

-	Practical Efficiency: Experiments on real-world datasets (e.g., cardiac MRI, hyperspectral imaging) show significant speedups (up to 100× faster than direct methods) while maintaining competitive reconstruction errors.

Weaknesses:

-	Clarity and Accessibility: some parts requires further clarification (see my detailed comments in the next section)

-	Comparison to State-of-the-Art: the paper does not benchmark against modern TC methods (e.g., tensor nuclear norm minimization, deep learning-based approaches), leaving its relative performance unclear.

**Questions For Authors:**

I have some questions that could help me to better understand the experimental results:

-	Why  does approximate-mini-als achieve lower error than other algorithms around step 15 in Fig. 1?

-	Could you please explain what do you mean the direct and parafac methods used in Fig. 2? You should precisely define those methods.

-	What is the difference between the plots in 3rd and 4th columns in Figure 2? They both compare the running time of the evaluated algorithms as a function of sample ratio. Do they refer to mini-ALS and accelerated-mini-ALS, respectively? Please, confirm.

**Relation To Broader Scientific Literature:**

The paper effectively integrates concepts from tensor decomposition, iterative optimization, and randomized numerical linear algebra. By leveraging preconditioned Richardson iteration and leverage score sampling, it advances the field of efficient tensor completion while maintaining theoretical guarantees.

However, it focuses solely on tensor decomposition-based completion, while many other tensor completion methods exist, including deep learning-based approaches and nuclear norm minimization, among others.

**Theoretical Claims:**

I have not verified the proofs provided in the Appendices. However, the theoretical claims appear well-founded.

---

> ### Author Rebuttal · Authors · 2025-03-31
>
> Thank you for your review and careful read. Please see our responses to your questions and weaknesses below.
>
> > The general TC problem of eq. (1) is not alligned with the approach of the current paper where a TD approach is used meaning that the approximation of the data tensor is a low-rank tensor decomposition (CP, Tucker or TT) where its factors are the minimizing parameter \theta. In other words, the regularization term of eq. (1) is not needed nor used in this paper.
>
> We agree that in the algorithm explicitly discussed in the paper, the regularization term of eq. (1) is not used. However, we emphasize that our approach can be applied to methods with regularization as well if the corresponding regression problems in such methods are structured and that structure enables faster algorithms. One example is Tucker decomposition with $\ell_2$-regularization, which reduces to solving *ridge regression* problems. For this problem, fast algorithms using sampling techniques are developed in [1], and our approach gives speed-ups for the corresponding tensor completion problem. Thank you for pointing this out — we will add a discussion about this to the next version.
>
> > The statement “Rank constraints can be incorporated into (1) by including appropriate penalty terms in R(\theta)” is confusing since in this paper, the low rank constraint is explicitly imposed by chosing the rank of the decomposition.
>
> We completely agree that the algorithms we explicitly describe in the paper do not incorporate rank constraints this way. The comment is meant to state that if there are regularization terms that still lead to structured problems that can be solved faster, then our lifting approach can give speed-ups for them as well, e.g., ridge regression that is discussed in the answer to the previous question. Note that the ridge regression reduces the effective dimension of the problem and can be used to implicitly induce a lower rank for the decomposition. We will make this clearer in the next version to avoid confusion.
>
> > In the paper, it is not clearly explained what the authors refers as direct and parafac methods.
>
> This is explained in the first column on page 8. The `direct` method uses normal equations for solving the linear regression problem in each iteration of the ALS algorithm. More specifically for a matrix $A$, vector $b$, and set of observed entries $\Omega$, we require to solve $x^*=\arg\min_{x} || A_{\Omega} x - b_{\Omega} ||_2^2$.
>
> The `direct` method computes $x^*$ using the following operation $x^* = (A_{\Omega}^{\top} A_{\Omega})^{-1} A_{\Omega}^{\top} b_{\Omega}$.
>
> The `parafac` method is the EM approach of Tomasi-Bro (2005), which is equivalent to running the mini-ALS algorithm for one step at each iteration of the ALS algorithm. We will add more detailed explanations about these algorithms in the next version of the paper.
>
> > What is the difference between the plots in 3rd and 4th columns in Figure 2? They both compare the running time of the evaluated algorithms as a function of sample ratio. Do they refer to mini-ALS and accelerated-mini-ALS, respectively? Please, confirm.
>
> Yes, the third column in Figure 2 shows the running times of the `mini-ALS` algorithm for different values of $\varepsilon$, while the fourth column shows the running times of the `approximate-mini-ALS` algorithm. This is explained in the text (lines 407–425, second column on page 8). We will add a comprehensive explanation to the caption of Figure 2 in the next version.
>
> > Figure 1 show that the approximate-mini-als algorithm achieves lower error than the other algorithms for a range of steps around step 15. This is surprising and should be explained.
>
> The `approximate-mini-ALS` in Figure 1 is a randomized method that uses leverage score sampling for the Kronecker product. We believe the lower error is due to randomness, which causes the algorithm to follow a different convergence path. We will provide an explanation of this in the next version of the paper.
>
> References:
>
> [1] Fahrbach, Matthew, Gang Fu, and Mehrdad Ghadiri. "Subquadratic kronecker regression with applications to tensor decomposition." Advances in Neural Information Processing Systems 35 (2022): 28776-28789.

---

> > ### Comment · Reviewer_NbQN · 2025-04-04
> >
> > I thank the author for acknowledgment of my constructive comments and their commitment to improve the explanations in the final version of the paper.
> > I keep my original recommendation

---

### Official Review · Reviewer_BWjF · 2025-03-10

**Overall Recommendation:** 4

**Summary:**

This paper addresses the problem of tensor completion (TC) by proposing a novel approach that leverages approximate Richardson iteration and structured tensor decomposition (TD) algorithms. The authors introduce a lifting technique to transform the TC problem into a structured linear regression problem, enabling the use of fast TD algorithms as blackbox subroutines. This approach allows for sublinear-time methods, significantly speeding up the completion process compared to traditional direct methods. The proposed method, termed approximate-mini-ALS, is theoretically analyzed for convergence and empirically validated on real-world tensors, demonstrating substantial speed improvements while maintaining comparable reconstruction accuracy.

**Claims And Evidence:**

The authors claim that their proposed method, approximate-mini-ALS, can achieve significant speedups in tensor completion tasks without compromising on the quality of the solution. The evidence supporting these claims includes:
1.	Theoretical Analysis: The authors provide a detailed convergence analysis of their approximate Richardson iteration-based algorithm, proving that it converges at the same rate as the exact Richardson iteration under certain conditions (Theorem 3.7).
2.	Empirical Validation: Extensive experiments on synthetic and real-world tensors (e.g., CARDIAC-MRI and HYPERSPECTRAL datasets) demonstrate that the proposed method can be orders of magnitude faster than direct methods while achieving similar reconstruction errors.
3.	Comparison with Existing Methods: The paper compares the proposed method with existing approaches such as direct ALS and the EM-based algorithm of Tomasi & Bro (2005), showing superior performance in terms of running time and solution quality.

**Essential References Not Discussed:**

No

**Experimental Designs Or Analyses:**

I think experiments are reasonably well designed. In addition, TC problem is often used for predicting missing values. The author states that the complexity of other algorithms is related to the number of observed values. I’m curious—when the number of observed values is small, does the author’s method have an advantage over traditional algorithms? Is it more efficient in terms of time, or does it have better recovery performance?

**Methods And Evaluation Criteria:**

The proposed method-approximate-mini-ALS using RRE to measure the relative Frobenius norm of the difference between the completed tensor and the ground truth.I think there is no problem.

**Other Comments Or Suggestions:**

•  x^{(k)} should represent the value from the k-th iteration of the ALS algorithm, but this is not clarified in the text. I believe this could confuse the readers, so please provide an explanation.
•  The author should provide the parameter settings used in the experiments or release the code to help readers better understand the work.

**Other Strengths And Weaknesses:**

The strength of this paper is that it proposes a novel and fast tensor completion algorithm with originality, and provides theoretical proofs to support it. Experimental results demonstrate its effectiveness. However, a weakness is that it remains unclear whether the proposed algorithm achieves faster speed or better recovery performance under high missing rate scenarios. Additionally, the author mentions tensor networks in the supplementary material—could their method be applied to tensor network algorithms, such as FCTN decomposition or TW decomposition?

**Questions For Authors:**

1. The author should provide the parameter settings used in the experiments or release the code to help readers better understand the work.
2. TC problem is often used for predicting missing values. The author states that the complexity of other algorithms is related to the number of observed values. I’m curious—when the number of observed values is small, does the author’s method have an advantage over traditional algorithms? Is it more efficient in terms of time, or does it have better recovery performance?
3. The author mentions tensor networks in the supplementary material—could their method be applied to tensor network algorithms, such as FCTN decomposition or TW decomposition?

**Relation To Broader Scientific Literature:**

The paper builds upon and contributes to several areas of the broader scientific literature:
1.	Tensor Completion: Extends the state-of-the-art in TC by proposing a novel lifting approach that leverages structured TD algorithms, addressing the limitations of existing methods that rely on direct matrix operations.
2.	Iterative Methods: Connects TC to the Richardson iteration, a classical iterative method for solving linear systems, providing new insights into the convergence properties of TC algorithms.
3.	Tensor Decompositions: Integrates with existing TD methods (e.g., CP, Tucker, tensor-train) by using them as subroutines, ensuring compatibility with a wide range of tensor structures and applications.

**Theoretical Claims:**

Yes, I check the proof of Lemma 3.1, 3.2, 3.5, Theorem 3.7. They are all correct.

---

> ### Author Rebuttal · Authors · 2025-03-31
>
> Thank you for your review, commenting on its novelty, and carefully checking the proofs of the main lemmas/theorems. Please see our responses to your questions and weaknesses below.
>
> > I’m curious—when the number of observed values is small, does the author’s method have an advantage over traditional algorithms? Is it more efficient in terms of time, or does it have better recovery performance?
>
> Please see the answer to the next question.
>
> > However, a weakness is that it remains unclear whether the proposed algorithm achieves faster speed or better recovery performance under high missing rate scenarios.
>
> The vanilla version of our algorithm can be slower than direct ALS methods when a small fraction of elements are observed, but our accelerated version gives significant speed-ups even for this case—see the 4th column of Figure 2. Note that this occurs when $\beta$ is large in Theorem 3.7, which happens if the lifted TD design matrix is not a tight spectral approximation to the TC design matrix. Our asymptotic running times imply that the speed-ups become even faster for larger tensors. In other words, for sufficiently large tensors with a small fraction of observed entries, our algorithm still provides significant speed-ups.
>
> > Additionally, the author mentions tensor networks in the supplementary material—could their method be applied to tensor network algorithms, such as FCTN decomposition or TW decomposition?
>
> Yes, we believe so. Recently, [1] has proposed a framework for designing sampling-based decomposition algorithms for arbitrary tensor networks. Therefore, we think someone could use that approach to design more efficient algorithms for FCTN decomposition and then, using our methods, extend it to tensor completion. Even more recently, sampling-based algorithms were designed for TW decompositions [2]. Our approach can extend their algorithm to tensor completion as well. We will add a discussion about these in the next version.
>
> > x^{(k)} should represent the value from the k-th iteration of the ALS algorithm, but this is not clarified in the text. I believe this could confuse the readers, so please provide an explanation.
>
> Thank you for the suggestion. We clarify this in the next version of the paper.
>
> > The author should provide the parameter settings used in the experiments or release the code to help readers better understand the work.
>
> Note that we included our code in the supplementary material. It is also available in a private GitHub repository. We will make the repository public and provide a link to it in the next version of the paper after the double-blind review process concludes.
>
> [1] Malik, Osman Asif, Vivek Bharadwaj, and Riley Murray. "Sampling-based decomposition algorithms for arbitrary tensor networks." arXiv preprint arXiv:2210.03828 (2022).
>
> [2] Wang, Mengyu, Yajie Yu, and Hanyu Li. "Randomized Tensor Wheel Decomposition." SIAM Journal on Scientific Computing 46, no. 3 (2024): A1714-A1746.

---

### Official Review · Reviewer_nT2U · 2025-03-14

**Overall Recommendation:** 3

**Summary:**

In this paper the well-known tensor completion problem for several popular low-rank tensor formats is considered. This problem is solved through a modified (randomized) ALS method, where it is assumed that we have access to all elements of the tensor (the so-called “lifting approach”). In those points whose values are not initially given, the values taken by the tensor at a given iteration is taken. Formally, the problem is mathematically written as a minimization problem not only on the elements of the expansion cores, but also on the unknown values of the tensor (the equivalence of these formulations is proved). One of the theoretical results of the paper is the construction of analogies of the obtained ALS iterations with Approximate Richardson Iteration. This allows authors to prove convergence theorems of the method. Small tensors (10^6~10^7 elements) are taken as numerical experiments and only the approximation time is compared with another methods (the accuracy of all methods is approximately the same).

**Claims And Evidence:**

The paper is largely theoretical, and it does have theorems for the convergence of the method and its running time. However, the experiments presented are rather synthetic and small, it is not clear where in real-world applications these techniques can be applied.

**Essential References Not Discussed:**

no

**Experimental Designs Or Analyses:**

Apart from the fundamental limitations of the experiments I wrote above, the numerical experiments in the paper are carried out without question.

**Methods And Evaluation Criteria:**

As numerical experiments, the authors take rather synthetic problems with tensors with a small number of elements. At the same time, the presented algorithm works well only when the ratio of the number of known elements to the number of unknown elements is rather large.
Disadvantages of this approach:
- Low-rank tensor decompositions were created to, among other things, overcome the “curse of dimensinality” --- situations where the full tensor does not fit in the memory of a computing system (or it is so large that it is difficult to work with).  When the elements of the tensor occupy a small memory (in the experiments in the paper --- at most tens of megabytes) the need for low-rank decompositions as such is small; and the problem of restoring missing elements can be solved in many other ways (e.g., spline interpolation).
- For the same reason, the case when a very small fraction of elements of all possible tensor elements is known is interesting.  In many real-world datasets this can be tenths and hundredths of a percent). And in this mode the presented algorithm performs poorly, as it can be seen from the plots.
- as a result, the authors show only the execution time of the algorithms (all of them have approximately the same accuracy). But for different algorithms, this time can strongly depend on the implementation, the use of hardware such as GPUs, the ability to parallelize and vectorize algorithms, etc. Thus, the only superiority of the presented algorithm in the experimental data, i.e., running time, is rather doubtful.

**Other Comments Or Suggestions:**

-

**Other Strengths And Weaknesses:**

Minor:
- l119: $:=R/R_k$ should not be in subscript

**Questions For Authors:**

- Have you tried your method on much larger datasets (including those for which the full tensor does not fit into computer memory)?

- which parafac implementation did you use in experiments in Fig.2?

**Relation To Broader Scientific Literature:**

In fact, this paper is a some amalgamation of several existing and developed ideas --- 1. The ALS method, which can be applied to all three tensor decompositions presented (and in each of them it can be written as a regression problem), 2. Score Sampling, see e.g. the work of , Bharadwaj et al. (2024,) 3. Richardson Iterations. But ussually Score Sampling is applied in the case where any element of the tensor is accessible, not to the tensor competion problem.

**Theoretical Claims:**

All theorems are proved correct in my opinion, theoretical results are true.

---

> ### Author Rebuttal · Authors · 2025-03-31
>
> Thank you for your review and for going over the theoretical results and experiments carefully. Please see our responses to your questions and weaknesses below.
>
> > The experiments presented are rather synthetic and small, it is not clear where in real-world applications these techniques can be applied
>
> Note that the `cardiac-mri` and `hyperspectral` tensors in Section 5.2 are real-world tensors, not synthetic.
>
> > Low-rank tensor decompositions were created to, among other things, overcome the “curse of dimensinality” --- situations where the full tensor does not fit in the memory of a computing system (or it is so large that it is difficult to work with). When the elements of the tensor occupy a small memory (in the experiments in the paper --- at most tens of megabytes) the need for low-rank decompositions as such is small; and the problem of restoring missing elements can be solved in many other ways (e.g., spline interpolation).
>
> We completely agree that this is a primary motivation for low-rank tensor decompositions. That said, another benefit of low-rank decomposition is to prevent overfitting because it can also be viewed as a form of regularization.
>
> In our experiments, our focus has been on comparison with baselines in terms of the running time and convergence. Our experiments demonstrate that we achieve a significant speed-up even for small tensors, and this speed-up only increases for larger tensors. Moreover, the memory footprint of our method is much smaller than the baselines due to the use of approximate solvers (using sampling techniques). In previous works on tensor decompositions that use leverage score sampling, it has been demonstrated that these methods are *especially effective* when direct methods run out of memory. Thus, we believe our experiments provide sufficient evidence for practicality and improvements of our algorithms.
>
> > For the same reason, the case when a very small fraction of elements of all possible tensor elements is known is interesting. In many real-world datasets this can be tenths and hundredths of a percent). And in this mode the presented algorithm performs poorly, as it can be seen from the plots.
>
> We disagree with this assessment. Although the vanilla version of our algorithm is not faster than direct ALS methods when a small fraction of elements are observed, the *accelerated version* gives significant speed-ups even for this case—see the 4th column (`accelerated-mini-als`) of Figure 2 and compare it with the 3rd column (`mini-als`). Our asymptotic running times (theorems in Section 4) imply that such speed-ups will be even faster for larger tensors. In other words, for sufficiently large tensors with a small fraction of observed entries, our algorithm still provides significant speed-ups.
>
> > But for different algorithms, this time can strongly depend on the implementation, the use of hardware such as GPUs, the ability to parallelize and vectorize algorithms, etc. Thus, the only superiority of the presented algorithm in the experimental data, i.e., running time, is rather doubtful.
>
> Our algorithms similarly benefit from the use of parallelizability, vectorized algorithms, and the use of GPUs since they’re iterative methods under the hood. Note that `numpy.linalg` (and hence `tensorly`) calls BLAS and LAPACK low-level operations, some of which are parallelized on CPU. Further, improvements in asymptotic running times of algorithms often lead to speed-ups that might not be achievable with a better code or hardware. Our approach unlocks such improvements for tensor completion via techniques that have been developed for tensor decomposition.
>
> > Have you tried your method on much larger datasets (including those for which the full tensor does not fit into computer memory)?
>
> We have not tried it on tensor datasets that exceed our machine’s memory. We believe this is a fantastic future direction of research, but it is somewhat out of the scope for this paper since such implementations require special care with respect to the code and hardware.
>
> > which parafac implementation did you use in experiments in Fig.2?
>
> PARAFAC refers to the EM approach of Tomasi-Bro (2005), which is equivalent to running our mini-ALS algorithm for exactly one step at each iteration of the ALS algorithm.

---

### Official Review · Reviewer_rkMi · 2025-03-23

**Overall Recommendation:** 3

**Summary:**

This paper introduces an efficient tensor completion algorithm, approximate-mini-ALS, which transforms unstructured tensor completion into a structured tensor decomposition problem using a lifting strategy and approximate Richardson iteration. By leveraging leverage score sampling, the method achieves sublinear time complexity and linear convergence with a contraction factor of \(1 - 1/\beta\), theoretically guaranteeing near-optimal sampling complexity \(O(n^{3/2})\) ().

**Claims And Evidence:**

Please refer to below

**Essential References Not Discussed:**

Please refer to below

**Experimental Designs Or Analyses:**

Please refer to below

**Methods And Evaluation Criteria:**

Please refer to below

**Other Comments Or Suggestions:**

There are some typos，eg: mode- -72 unfolding...
The author should check typos carefully.

**Other Strengths And Weaknesses:**

Strengths：
1.Innovative Combination: The lifting strategy combined with approximate Richardson iteration and leverage score sampling offers a novel approach to tensor completion, leveraging existing techniques in a non-trivial way.
 2. Compared with ALS algorithms, the algorithm speed is improved significantly.
Weaknesses:
1.In the experimental section, the author presents too few experimental results, which do not provide sufficient evidence to convincingly demonstrate the advantages of the proposed algorithm. The author should refer to article and increase the number of experiments.

**Questions For Authors:**

1.The author should provide ablation experiments.
2.Whether the method can be transferred to t-svd framework？

**Relation To Broader Scientific Literature:**

Please refer to below

**Theoretical Claims:**

Please refer to below

---

> ### Author Rebuttal · Authors · 2025-03-31
>
> Thank you for your review and for recognizing that our approach is an innovative combination. Please see our responses to your questions and weaknesses below.
>
> > In the experimental section, the author presents too few experimental results, which do not provide sufficient evidence to convincingly demonstrate the advantages of the proposed algorithm. The author should refer to article and increase the number of experiments.
>
> We believe our experiments provide sufficient evidence for the speed-up of our algorithms in practice (see the running time subplots in Figures 2 and 3). Please note that due to the improved asymptotic running time of our algorithm (several theorems in Section 4), the speed-up of our algorithm (relative to baselines) only gets faster for larger tensors. That said, we appreciate the suggestions for extra experiments and are curious which “article” you are referring to.
>
> > The author should provide ablation experiments
>
> We believe our experiments include an ablation study, as we have provided experiments (1) with and without acceleration (i.e., the fourth column vs the third column in Figure 2; `accelerated-mini-als` is faster than `mini-als`), and (2) with and without leverage score sampling (i.e., `approximate-mini-als` vs `mini-als` in Figure 1). We appreciate more specific suggestions if there are ablations that you think would improve the paper.
>
> > Whether the method can be transferred to t-svd framework？
>
> Some recent studies have considered more efficient algorithms for t-SVD using leverage score sampling—see [1]. Due to this connection, we believe it might be possible to adopt our method for tensor completion under the t-SVD framework as well. We will add a more detailed discussion about this to the next version of the paper.
>
>
> [1] Tarzanagh, Davoud Ataee, and George Michailidis. "Fast randomized algorithms for t-product based tensor operations and decompositions with applications to imaging data." SIAM Journal on Imaging Sciences 11, no. 4 (2018): 2629-2664.

---

### Decision · Program_Chairs · 2025-05-01

**Decision:**

Accept (poster)

**Comment:**

The paper reviewers all agree that the paper is worth publishing at ICML. The combination of methods in the paper seems innovative and the paper seems like a good contribution.